# Human transcription factor protein interaction networks

Helka Göös [1,4], Matias Kinnunen[1], Kari Salokas [1,5], Zenglai Tan[2,5], Xiaonan Liu [1], Leena Yadav[1], Qin Zhang[2], Gong-Hong Wei[2,3] & Markku Varjosalo [1✉]

Transcription factors (TFs) interact with several other proteins in the process of transcriptional regulation. Here, we identify 6703 and 1536 protein–protein interactions for 109 different human TFs through proximity-dependent biotinylation (BioID) and affinity purification mass spectrometry (AP-MS), respectively. The BioID analysis identifies more high-confidence interactions, highlighting the transient and dynamic nature of many of the TF interactions. By performing clustering and correlation analyses, we identify subgroups of TFs associated with specific biological functions, such as RNA splicing or chromatin remodeling. We also observe 202 TF-TF interactions, of which 118 are interactions with nuclear factor 1 (NFI) family members, indicating uncharacterized cross-talk between NFI signaling and other TF signaling pathways. Moreover, TF interactions with basal transcription machinery are mainly observed through TFIID and SAGA complexes. This study provides a rich resource of human TF interactions and also act as a starting point for future studies aimed at understanding TF-mediated transcription.

[1] Institute of Biotechnology, HiLIFE, University of Helsinki, Helsinki, Finland. [2] Biocenter Oulu and Faculty of Biochemistry and Molecular Medicine, University of Oulu, Oulu, Finland. [3] Department of Biochemistry and Molecular Biology, School of Basic Medical Sciences, Fudan University Shanghai Cancer Center, Shanghai Medical College of Fudan University, Shanghai, China. [4] Present address: Finnish Red Cross Blood Service, Helsinki, Finland. [5] These authors contributed equally: Kari Salokas, Zenglai Tan. ✉email: markku.varjosalo@helsinki.fi

'The central dogma' states that genetic sequence information from DNA is transcribed to RNA and subsequently translated into proteins. These processes are tightly regulated and employ a plethora of proteins. Transcription, the first step, is regulated by transcription factors (TFs), which represent one of the largest families of human genes. In humans, 6–9% (~1400–1900) of proteins are predicted to regulate gene expression through DNA binding (refs. [1–3], https://www.proteinatlas.org), and the most recent manual curation identified 1639 likely human TFs[4].

Complex and multilayer regulation of transcription involves not only direct binding of TFs to a target gene's regulatory element(s) but also a complicated interplay between TFs and TF binding proteins. These include several cofactors, the Mediator complex, basal transcription machinery, TF activity modulating enzymes (such as phosphatases and kinases), dimerization partners, subunits and inhibitory proteins[5–8]. Moreover, as chromosomal DNA is packed into chromatin to prevent uncontrolled transcription, TFs also interact with several chromatin remodeling proteins. The formed complexes are necessary to regulate the accessibility of DNA to allow chromatin opening and thereby gene transcription.

TFs play crucial roles in regulating numerous cellular mechanisms and are key regulators of tissue growth and embryonic development – processes that may cause cancer and other disorders when aberrantly controlled. Therefore, understanding the TF network at the systems level would build an important foundation for future studies as well as for therapeutic approaches[7]. While the binding of TFs to DNA is relatively well studied, for the most part, we still lack a global understanding of TF protein–protein interactions (PPIs) and their roles in the regulation of transcription. Therefore, we sought to fill this knowledge gap by using recently developed state-of-the-art PPI identification methods, which allow unprecedented sensitivity and depth of analysis.

In this study, we systemically characterized the PPIs of a selected set of 109 human TFs using affinity purification mass spectrometry (AP-MS) and proximity-dependent biotinylation (BioID) mass spectrometry. We identified 6703 PPIs in the BioID analysis and 1536 PPIs in the AP-MS analysis. Most of the detected interactions were contextually nuclear and linked to transcription and transcriptional regulation. These interactions paint a picture of how transcription factors are activated or repressed and add experimental evidence for the potential relevance of transient interactions in the advent of transcription-related nuclear condensates and phase separation. This large interactome network of TFs allowed us to recognize several interactome subgroups of TFs, such as TFs linked to mRNA splicing and TFs linked to chromatin remodeling. In addition, we observed that most of the studied TFs interacted with nuclear factor 1 (NFI) TFs, which are essential for several developmental and oncogenic processes. Overall, this work represents a rich resource to direct future studies aimed at understanding TF-mediated transcription and how TF-formed interactions regulate important cellular phenomena in both health and disease.

## Results

**Identification of TF protein–protein interactions.** To systematically investigate the protein–protein interactions of human TFs, we selected a representative set of 109 TF genes from different TF families (Supplementary Data 1a). Selection was based on the availability of full-length TF constructs. Selected TFs were analyzed in two biological replicates and, as the correlation between the technical and biological replicates were excellent (Supplementary Fig. 1a), either in one or two technical replicates.

TFs are often classified according to their DNA-binding domains (DBDs), and the DBD distribution of studied TFs compared to all human TFs is shown in Fig. 1a. The majority of the studied TFs had C2H2 zing finger (ZF) or homeodomain DBDs, which are the most common DBDs among the human TFs[4].

The selected TFs were subjected to two independent mass spectrometry-based interactome analysis methods (Fig. 1b). First, the stable TF complexes were purified using single-step Strep-tag affinity purification (AP-MS). Second, a proximity-dependent labeling approach (BioID) utilizing a minimal biotin ligase (BirA*)-tag was used to detect transient and proximal interactions of the TFs[9,10]. Activation of BirA allows it to biotinylate proteins within close proximity (10 nm) of the studied TFs, including transient interactions. However, no physical contact is needed for biotinylation, and because of the confined nature of chromatin, proteins other than interacting proteins might also be labeled in low amounts. The expression of the studied TFs was adjusted on the corresponding transgenic cell lines by the tetracycline inducible and adjustable Tet-On expression system[11] – resulting in expression levels from close-to-physiological levels to mild or moderate overexpression.

In total, we identified 6703 high-confidence PPIs using BioID analysis and 1536 PPIs using the AP-MS method (Supplementary Data 1b, c and Fig. 1c–e). Of these, 200 were detected with both methods (Fig. 1c). For an initial quality check for the obtained TF interactomes, we mapped the interactors to their known subcellular localizations from the Cell Atlas[12]. This analysis revealed that 80% of the TF interactors were nuclear localized (yellow nodes; Fig. 1d and Supplementary Data 2), confirming the expected nuclear compartmentalization of the studied TFs and their interactors. Remarkably, the majority (>75%) of the interactions within the TF interactome were previously unreported (Fig. 1e and Supplementary Data 1b, c). On average, we identified 66 PPIs/TF in the BioID data and 16 PPIs/TF in the AP-MS (Fig. 1f). The higher number of identified interactions by BioID compared to AP-MS agreed with many recently reported medium- and large-scale interactomics studies[13–16]; however, our results strongly suggest that TFs prefer to form more transient or proximal interactions than stable protein complexes. This finding is consistent with the phase separation model for TF interactions, where interactions incorporated in TF condensates are dynamic[17–20].

It has been suggested that the BioID method is efficient for studying transient interactions[9], and this was supported by our results, which strongly suggested that BioID is the method of choice for studying TF interactions (Figs. 1e and 2a). Most of the TFs showed more detected high-confidence interactions with the BioID method, with only a few exceptions of Krüppel-like factor (KLF) family of transcription factors (Fig. 1f). There were prominent differences in the number of detected PPIs between different TF families; for example, SPs, TLXs, HNFs, and PAXs had over 100 PPIs on average, whereas NFACs, IRFs, STATs, GLIs, ETVs, and TEADs had fewer than 50 PPIs on average (Fig. 2b and Supplementary Data 3).

The most common TF interactor observed in our study was lysine-specific demethylase 2B (KDM2B), which interacted with 62 TFs (Supplementary Data 1b). In addition, two lysine methyltransferases were among the top five of the most frequent TF preys (KMT2D: 58 PPIs and KDM6A: 53 PPIs), which highlights the importance of histone modification homeostasis in the regulation of transcription. The detected interactions of lysine methyltransferases with TFs are highly specific and very rarely detected in large-scale studies with other key cellular signaling molecules[16,21,22] and hardly ever detected as contaminants[23]. Other common TF interactors were NFIA (54 PPIs), TLE1 (53

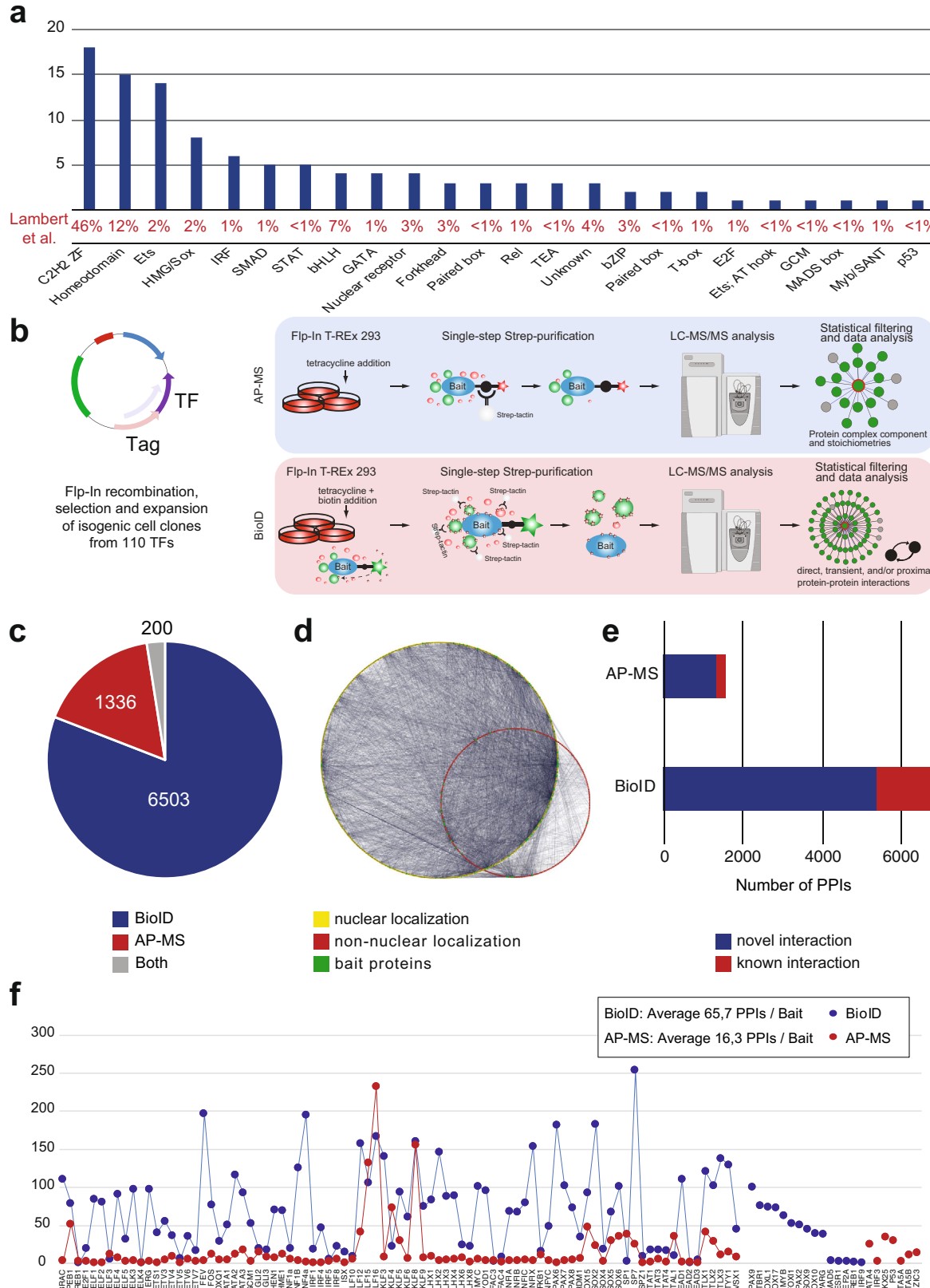

PPIs), CIC (52 PPIs), and several zinc-finger proteins (50–52 PPIs). In addition, the well-established corepressors BCOR (48 PPIs) and NCOR2 (48 PPIs) were high on the list. Not surprisingly, the most frequently observed TF interactors were transcriptional activators and repressors.

To obtain a glimpse to the biological nature of TF interactions, we performed Gene Ontology biological process (GO-BP) enrichment analysis for all BioID interactions (Fig. 3a, Supplementary Data 4). As expected, BP terms linked to transcription and its regulation were significantly enriched. The most

**Fig. 1 TF protein interactome identified using the BioID and AP-MS methods. a** The distribution of the DNA-binding domains of the studied TFs. The corresponding proportion of each DNA-binding domain from 1639 TFs in the study of Lambert et al. is shown as a percentage value below the graph. **b** Schematic illustration of the analysis methods used to comprehensively map the physical and functional interactions formed by the TFs. The TFs were tagged N-terminally with MAC, StrepIII-HA or BirA -tags (Supplementary Data 1a) and cotransfected with Flp-In recombinase to generate stable isogenic and inducible cell lines. Cells were induced by tetracycline addition for the corresponding TF expression and for the BioID analysis supplemented with biotin for 24 h. This was followed by cell harvesting, lysis, and affinity purification with Strep-beads. Purified proteins were further digested into peptides and analyzed by LC–MS/MS. Proteins were later identified, quantified, and analyzed to distill the high-confidence interactors using different statistical and bioinformatic methods. **c** A total of 6503 high-confidence protein–protein interactions were detected only with the BioID method, 1336 with the AP-MS method, and 200 with both the BioID and AP-MS methods. **d** Localization of interacting prey-proteins from BioID data according to the annotated localizations of Cell Atlas[12]. Yellow nodes (large circle) indicate nuclear localization, and red (small circle) indicates nonnuclear localization. Of the mapped proteins, >80% had nuclear localization. **e** Protein–protein interactions were identified using the AP-MS (1536) and BioID (6703) methods. Interactions were compared to interactions from the PINA2, IntAct, BioGRID, and String experimental protein interaction databases and to interactions from a study by Li et al.[6], by Lambert et al.[25], Malovannaya et al.[26], and Huttlin et al.[27] resulting in 220 and 1316 previously reported interactions in the AP-MS and BioID data, respectively. The proportions of known interactions are shown in red. **f** Number of high-confidence protein–protein interactions of different TF baits detected by AP-MS (red) or BioID (blue) affinity purification combined with mass spectrometry. Source data of the figure are provided as a Source Data file.

significantly enriched term was 'transcription, DNA-templated', with a $p$ value of $9.15 \times 10^{-97}$. This was followed by biological processes linked to positive and negative regulation of transcription with $p$ values of $<2.17 \times 10^{-31}$.

**Comparison to other studies**. As BioID can capture transient and proximal interactions, most experimental validation methods, such as coimmunoprecipitation, might not be sensitive enough for validation of the results. We, therefore, first compared the identified PPIs with previously published interactions. In a medium-scale analysis, Li et al. screened the PPIs of 59 TFs by tagging them with an SFB-tag (S protein-tag, FLAG-tag, and streptavidin binding peptide) and identified the interacting proteins using double affinity purification followed by MS analysis[6]. In this analysis, they identified 2156 PPIs. Fourteen of the TFs analyzed in their study were included in our set (CREB1, ETS1, FOS, FOXI1, FOXL1, FOXQ1, IRF3, MEF2A, MYC, NFKB1, PPARG, STAT3, TEAD2, and TP53). The double affinity purification method is not as sensitive as our single-step AP-MS or BioID approaches. Therefore, it is not surprising that the overlap between our BioID PPIs and their PPIs was low; only 5% of our PPIs were covered by their study (Supplementary Data 1b). A comparison with our AP-MS results revealed more common interactions; 25% of our AP-MS PPIs were detected by their approach (Supplementary Data 1c). These differences detected between Li et al.'s PPIs and those we identified are most likely due to the transient nature of TF interactions and the use of different tagging strategies.

Next, we compared our PPIs to public interaction databases such as PINA2[24], STRING, IntAct, and BioGRID and several medium- to large-scale interactome studies such as Lambert et al.[25], Malovannaya et al.[26], and Huttlin et al.[27] (Tables S1B, C). Overall, 20% (1316/6703) of our BioID PPIs and 14% (220/1536) of our AP-MS interactions were also found in public databases or in the abovementioned interatomic studies. The PPIs of several well-studied TFs, such as SOX2, MYC, TYY1, PAX6, HNF4a, and GATA2, overlapped with more than 45 known interactions in the databases, whereas the PPIs of other less studied TFs, such as ZIC3, ELK4, ESR1, IRF3, and IRF9, did not overlap with any known PPIs from the databases or studies (Supplementary Data 1b). While the overlap of identified interactions between our dataset and the previously mentioned large-scale interactomes studies is low, so is the overlap among the previous studies as well. Indeed, our results overlap more with the existing data, than many of the other interactomes (Supplementary Fig. 1b). This may suggest that all of the studies capture different facets of TF

interactomes, and together form a much more complete picture, than any one study alone.

**Clustering of transcription factor protein–protein interactions**. To study whether the identified PPIs of the various TFs correlated with their DBD families, we performed hierarchical clustering of baits by their prey intensities and compared that to bait DBDs. Only a modest correlation was seen between the PPIs and DBDs: TLX and LHX homeodomain TFs and KLFs and TYY1-C2H2 ZF TFs clustered together, but no other correlations with DBDs were observed (Fig. 3b).

Next, we determined whether PPI clustering correlated with TF amino acid sequences. To accomplish this, we aligned the full amino acid sequences and compared them to hierarchical PPI clustering (Supplementary Fig. 1c). The sequence alignment comparison to PPI clustering revealed multiple clusters with similarities in PPIs and sequences (Supplementary Fig. 1c), including the clusters of ELFs, NFIs, LHXs, and KLFs.

In addition, the DNA-binding motifs of the studied TFs were aligned using the matrix-clustering tool RSAT (Supplementary Fig. 1d)[28].

**TF interactions with basal transcription machinery and the Mediator complex**. Eukaryotic gene transcription is mostly executed by RNA polymerase II (Pol-II), which binds to conserved core promoters. In addition to Pol-II, the core promoters also bind the SAGA complex and the basal transcription machinery (also known as the preinitiation complex, PIC), which is composed of Pol-II, Mediator complex, and general TFs (GTFs). The GTFs are TATA-binding protein (TBP), TFIIA, TFIIB, TFIID, TFIIE, TFIIF, and TFIIH (Supplementary Data 5; refs. [29,30]). To assess how the studied TFs interacted with PIC components, we retrieved GTFs and Mediator complex members from the CORUM protein complex database and compared them to the identified PPIs (Supplementary Data 5). We observed multiple interactions with both TFIID and SAGA complex components but only a few interactions with the Mediator complex members (Supplementary Data 5), and we did not detect interactions between the studied TFs and TFIIA, TFIIB, TFIIF, TFIIH, or Pol-II complex components. This indicated that under the given conditions, the TF activity from enhancers to the core promoter and PIC is mainly mediated by TFIID, SAGA, and Mediator complexes.

**TF interactions with nuclear factors**. Interestingly, we found 202 TF-TF (bait-bait) interactions in our TF interactome (Fig. 4a).

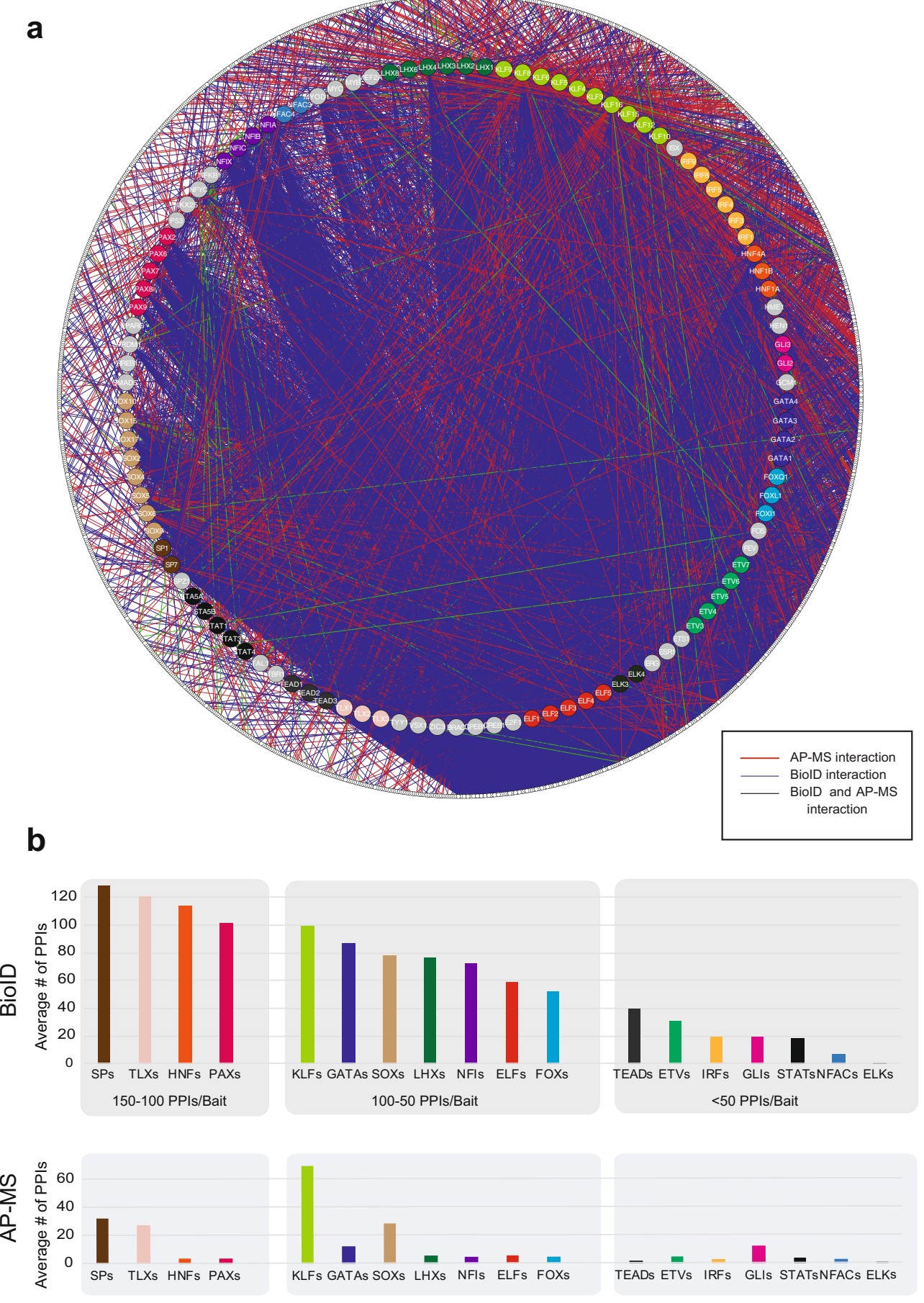

**Fig. 2 Comprehensive protein interactomes of the studied TF and TF families. a** Studied TFs are organized and color-coded (node color) based on their TF families in the inner circle, and interacting proteins are shown in the outer circle with white. Blue edges indicate interactions detected with the BioID analysis, red with the AP-MS analysis and green from both. **b** The average number of PPIs of different TF families detected by BioID and AP-MS. Note that the color coding of the TF families is the same as in a) and the TF families are organized into three bins based on the BioID data, first having families with 150-100 high-confidence interactions/bait TF, second 100-50, and third <50 interactions/TF. The stable interactions detected with AP-MS showed a correlation with the interaction numbers; however, the KLF family behaved differently, with a more than 2-fold higher number of average interactions than any other TF family. Source data of the figure are provided as a Source Data file.

The majority of these interactions (118) were TF interactions with NFI family members (NFIA, NFIB, NFIC, and NFIX; Fig. 4b). A total of 55 TFs interacted with one or more NFIs (Fig. 4b): 53 TFs interacted with NFIA, 37 with NFIB, 14 with NFIC, and 5 with NFIX (Fig. 4b). In addition, all four NFIs formed bidirectional interactions (when used reciprocally as bait proteins) with each other in both the AP-MS and BioID analyses (Fig. 4b). NFIs also had multiple other shared interactions than the abovementioned interactions with TFs (Supplementary Fig. 2a).

Of all eight studied SOX family members, only SOX4 had no interactions with NFI proteins (Fig. 4a). This suggested a previously uncharacterized crosstalk between SOX transcription factors and NFI signaling. Moreover, we found that all five studied PAXs interacted with NFIs (Figs. 4b and S2b). In fact, PAX9 was the only TF in the set that interacted with all four NFIs. To our knowledge, no link between PAX9 and NFIs has been reported before. In addition to SOXs and PAXs, LHXs (four out of six), KLFs (6/10) and all three studied TLXs interacted with NFIs (Fig. 4b). These and the other detected TF-NFI interactions (Fig. 4b) indicated that NFIs take part in multiple cellular processes with other TFs, possibly regulating their activity and functions. GO-BP enrichment analysis of NFI interactomes using the total BioID interactome as a background showed that transcription-related BP terms were significantly enriched, indicating the importance of NFIs in transcription regulation in general (Supplementary Data 4).

To validate the TF interactions with NFIs, we used coimmunoprecipitation (Co-IP) dot blot (DB) analysis. In this analysis, 37 identified TFs with interactions with NFIA, NFIB, and/or NFIC (Supplementary Fig. 3a, b) were tagged with the V5 epitope and coexpressed with NFIs. All 37 TFs were assayed with all three NFI family members, including combinations that were not detected in the mass spectrometry interactomics analyses. The Co-IP DB assays validated 65 out of 70 (93%) tested MS-identified interactions. The Co-IP assay also detected 33 new interactions that were not detected by mass spectrometry. FOS, FOXl1, KLF8, KLF16, LHX3, PRDM1, SOX9, and TYY1 interacted with NFIA, NFIB, or NFIC in the mass spectrometry results but interacted with all three in the Co-IP DB assays. BRAC, ELK3, ERG, ETV7, GATA3, HME1, KLF3, LHX2, LHX4, PAX2, PAX6, PAX8, SOX10, SOX15, SOX17, and TLX2 interacted with two of the NFIA, NFIB, or NFIC in the mass spectrometry results and with all three in the Co-IP DB assays. Of all tested interactions, only 5 interactions seen by MS, could be classified as negative. The interactions of NFIA with ELF1, ELF2, FOXL1, and KL12 and NFIB with KLF12 could not be detected by coimmunoprecipitation, and the interaction between NFIA and HEN1 was barely detectable, possibly suggesting that these interactions are not direct but mediated with other proteins, or are extremely transient in their nature.

Given that in our analyses NFIA interacted with 55 TFs, it is possible that this interaction with NFIA could regulate the transcriptional activity of other TFs. To test this hypothesis, we generated luciferase-based reporter (DNA binding domains extracted from JASPAR[31]) assays for selected TFs interacting with NFIA. Of note is that each of these assays measure only the possible binding activity change of each specific TF. TFs bind to several different sites in the DNA[32] and it is highly unlikely that DNA mediated interactions would be detected in these abundancies as detected by our interactome analyses.

Reporter assays that displayed clear activity change after the introduction of the corresponding TF were chosen for NFIA siRNA-mediated knockdown experiments. Knockdown efficiency was first confirmed by western blotting using a specific antibody against NFIA (Fig. 4c). Next, the effect of NFIA depletion on selected reporter activity was tested in the presence and absence of the corresponding TF (Fig. 4d). Interestingly, both KLF4 assays, which detected the repressive and activating response of KLF4, showed altered luciferase activity after NFIA silencing: both the repressive and activating responses after KLF4 induction were reduced (Fig. 4d). In addition, SOX2 and PAX6 showed reduced activity after NFIA silencing, while HME1 activity was increased after the depletion of NFIA (Fig. 4d).

**Prey-prey correlation analysis reveals several biological clusters.** The TF prey-prey correlation analysis using ProHits-Viz[33] revealed 17 biological clusters (Fig. 5 and Supplementary Data 6). This analysis revealed clusters of preys that were often seen together between baits, suggesting that they might be part of the same complex or molecular context or both. Baits driving the same cluster had a similarity in interactomes, indicating possible shared or similar biological roles. The preys belonging to different clusters and the baits driving the clusters are shown in Supplementary Data 6. Next, we describe some of the interesting clusters found in this correlation analysis.

**TF interactions with chromatin-modulating complexes in Cluster K.** Preys in Cluster K (Figs. 5 and 6a and Supplementary Data 6) had clear biological roles in chromatin modulation, especially in histone H4 and H3 modifications. Of these 100 preys, 67 were directly linked to histone and chromatin signaling (Fig. 6a and Supplementary Data 6). These included 10 members and one putative regulator of the INO80 chromatin remodeling complex (Fig. 6a), 7 members of the nonspecific lethal (NSL) complex, and 14 members of the MLL1-WDR5 histone-3-lysine-4-(H3K4) methyltransferase complex.

Cluster K mainly consisted of TYY1 interactions; TYY1 interacted with 99 preys in the cluster (Fig. 6a and Supplementary Data 6), while HNF4A interacted with 53, ELF4 with 48, ELF1 with 45, KLF8 with 40, and MYC with 39 preys. Other baits driving the cluster are listed in Supplementary Data 6.

TYY1 is known to be part of the INO80 complex[34]. As predicted, we found TYY1 to interact with 8 subunits of the INO80 complex and with its putative regulator UCHL (Fig. 6a; Supplementary Data 6). TYY1 interactions with INO80 complex members appear to be very stable, as many of the interactions were also detected in the AP-MS data (Supplementary Data 1c). In addition to TYY1, we found that ELF4 interacted with six INO80 complex members and UCHL (Fig. 6a and Supplementary Data 1b).

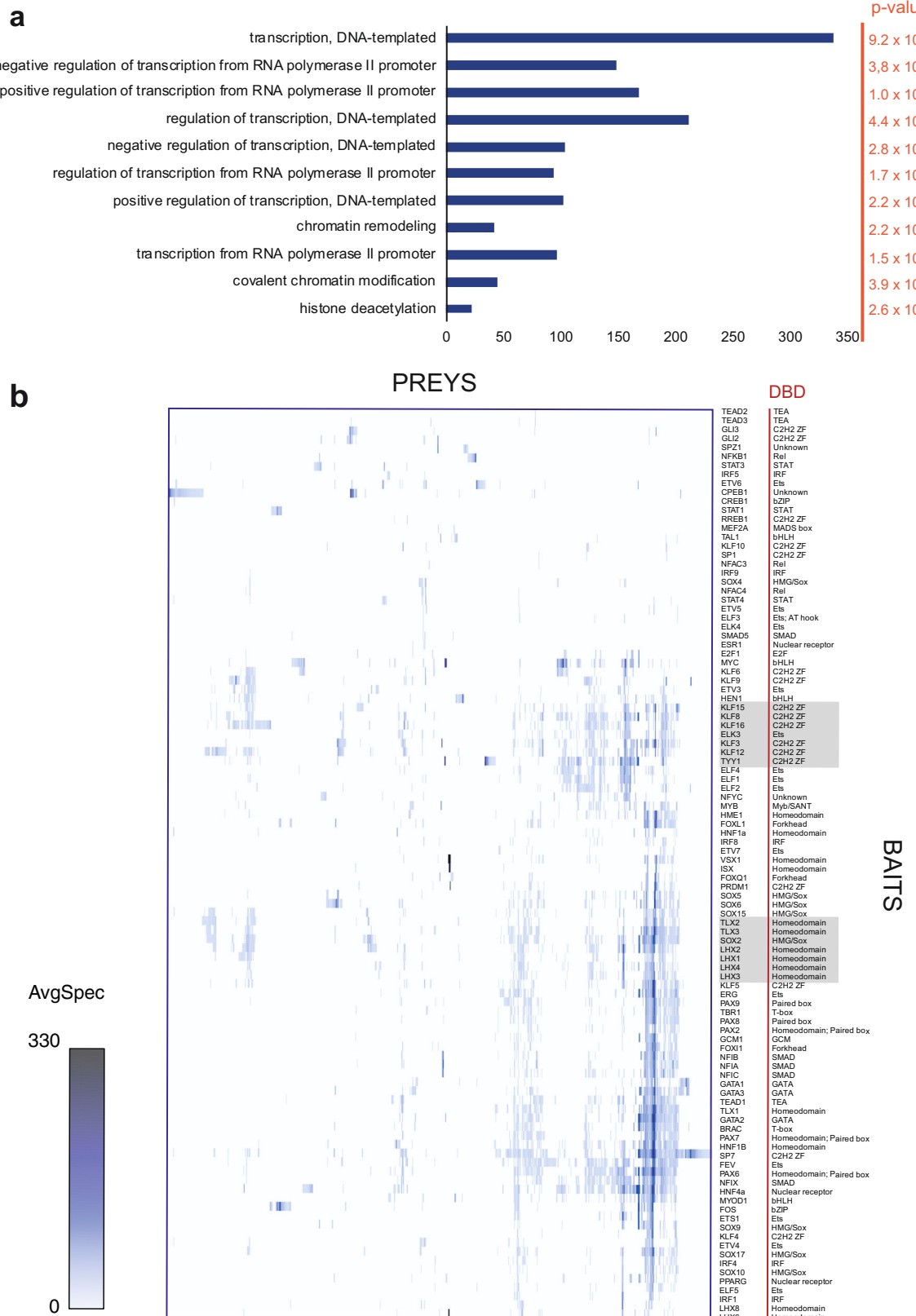

**Fig. 3 Hierarchical clustering of baits by preys and its correlations to DNA-binding domains of studied TFs. a** Enriched Gene Ontology Biological Processes of the high-confidence interacting proteins from the BioID analysis indicate high enrichment of DNA-templated transcription regulating proteins and chromatin modifiers as TF interactors. **b** The TF bait proteins and their interacting prey proteins from BioID analysis were hierarchically clustered (ProHits-viz.) and visualized in the heatmap (note that the color gradient shows the interaction abundancy (spectral counts) value for the prey). Source data of the figure are provided as a Source Data file.

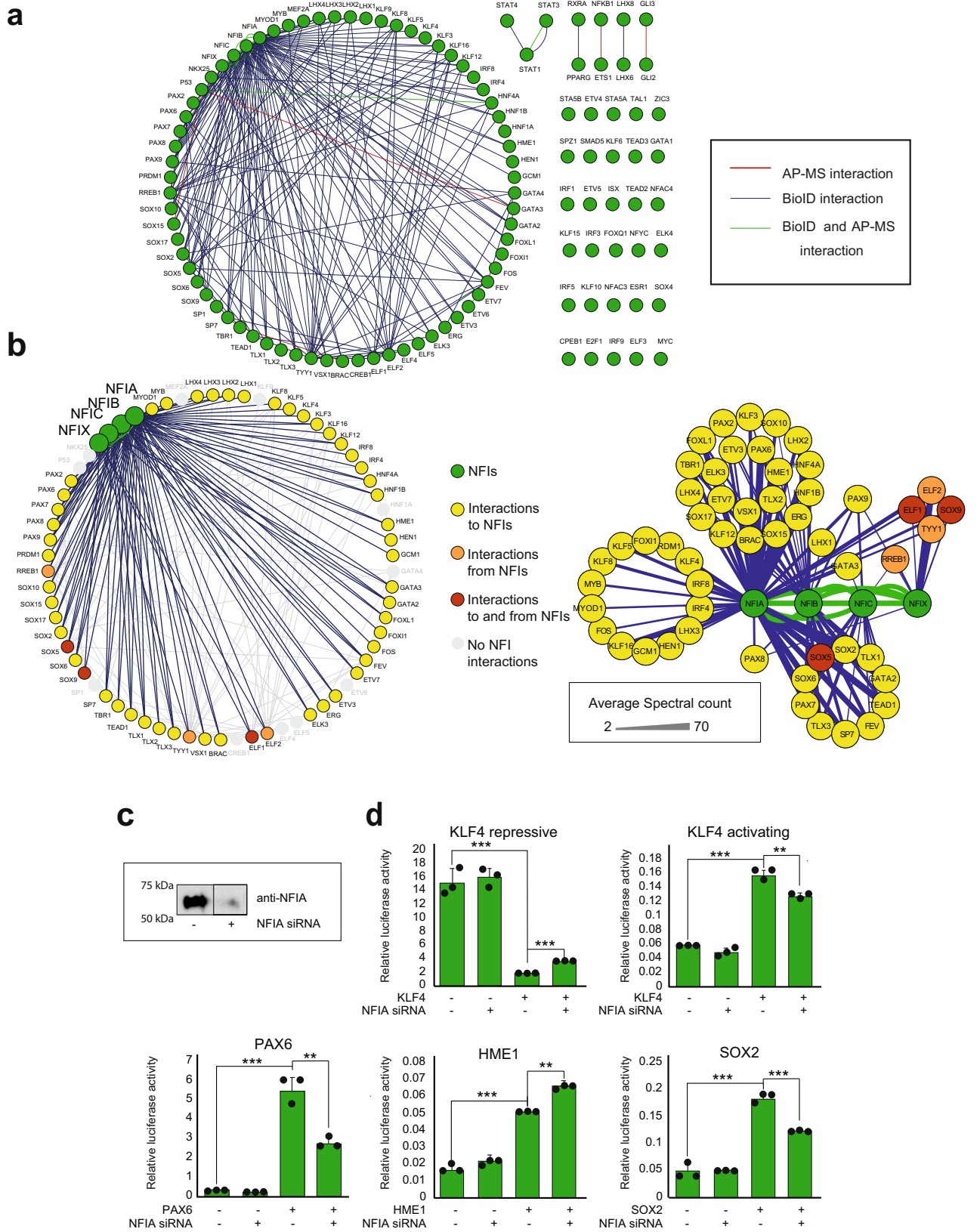

Cluster K also contained eight out of nine members of the NSL histone acetyltransferase complex (Fig. 6a and Supplementary Data 1b). In total, we found that TYY1 and MYC interacted with all eight identified subunits of the NSL complex, whereas HNF4A interacted with six subunits, and ELF1, ELF2, and ELF4 interacted with five subunits (Fig. 6a and Supplementary Data 6). Histone acetyltransferase KAT8 (main unit of the NSL complex, also known as MYST1) was found to interact with TYY1, ELF1, ELF2, ELF4, HNF4a, and MYC.

**Fig. 4 TF-TF (bait-bait) interactions of 109 TFs studied. a** Of 109 studied TFs, 80 had 202 interactions with other studied TFs. Blue edges indicate interactions from the BioID analysis, red from the AP-MS analysis and green from both. **b** Most of these TF-TF (118) interactions were TF interactions with NFIs (left panel). The right panel shows the separate groups shared by one or multiple NFIs. Color code: Green nodes = NFIs, yellow nodes = interactions to NFIs, orange nodes = interactions from NFIs, red nodes = interactions to and from NFIs and gray nodes = no interactions to or from NFIs. Color coding of the nodes is shown on the right side of the figure. The edge weight displays the average spectral count value detected for the corresponding interaction. **c** NFIA was silenced using siRNA transfection, and NFIA levels were detected 48 h after transfection by western blotting using a specific antibody against NFIA. **d** TF activity was investigated after NFIA silencing using both repressive and activating reporter gene analysis. Both the repressing and activating functions of KLF4 were reduced upon NFIA silencing. In addition, SOX2 and PAX6 activity was reduced, while HME1 activity was increased upon NFIA silencing. The representative data from two repeated experiments are presented as mean values +SD as appropriate. Two-sided $t$-test was used. $N = 3$, ***$p < 0.001$, **$p < 0.01$, *$p < 0.05$. Source data of the figure are provided as a Source Data file.

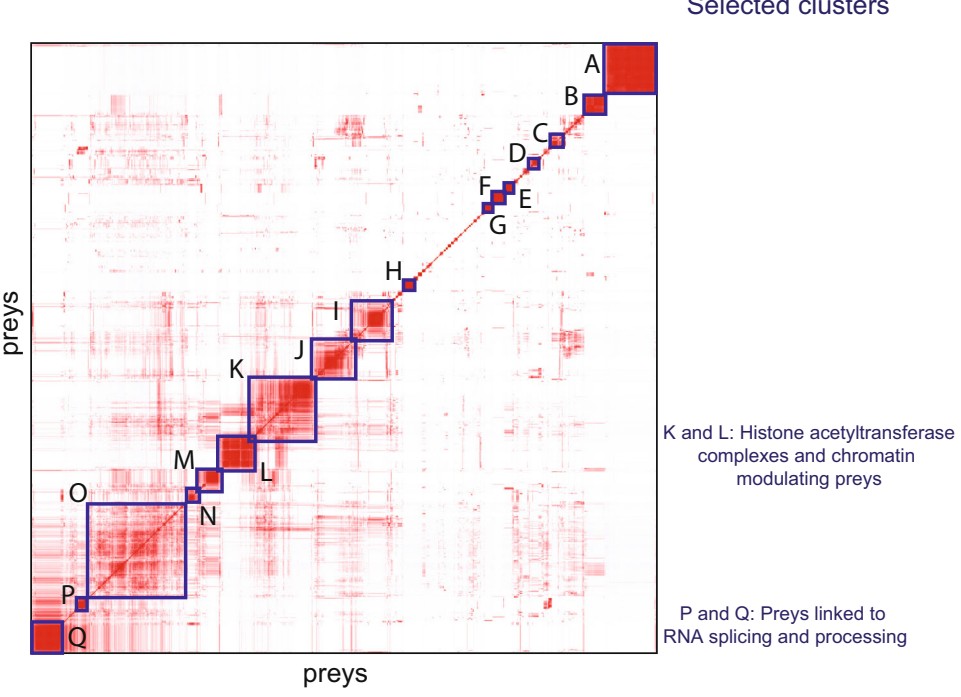

**Fig. 5 Biological clusters from prey-prey correlation analysis.** Prey-prey correlation analysis (ProHits-viz.) of prey identified in the BioID experiments. The results are shown in the heatmap with prey on both the x- and y-axes. Prey in clusters (A-Q) and baits driving the clusters are shown in Supplementary Data 6. The color scale indicates the Pearson correlation of prey PSMs that was performed by Prohits-viz. prior to hierarchical clustering. Source data of the figure are provided as a Source Data file.

**TF interactions with histone acetyltransferase complexes in Cluster L.** Closer analysis of Cluster L (Fig. 5) revealed accessory chromatin-modulating complexes, especially histone acetylation complexes. GO-BP analysis revealed a significant enrichment of terms linked to histone H2A, histone H4 and histone H3 acetylation ($p$ values of $<6.8 \times 10^{-26}$).

In total, Cluster L consisted of 53 preys, of which 40 were directly linked to histone modifications (Fig. 6a and Supplementary Data 6). These included 18 of 19 members of the SAGA complex and 14 members of the NuA4/Tip60 HAT complex A (Fig. 6a). The cluster was mainly driven by MYC, which had interactions with all 54 preys. KLF6 interacted with 30, HNF4A with 24, KLF8 with 22, ELF4 with 19, TYY1 with 14 and ELF1 with 12 preys. Other baits are listed in Supplementary Data 6.

MYC interacted with all 18 subunits of the SAGA complex identified in our data (Supplementary Data 6). Furthermore, KLF6 was found to interact with 15, KLF8 with eight, and ELF4 and HNF4a with seven SAGA complex subunits (Supplementary Data 6).

In addition, 14 of 15 NuA4/Tip60 HAT complex A subunits were identified in Cluster L. MYC was found to interact with all 14 identified subunits, while HNF4a and KLF6 interacted with

nine subunits, and KLF8 and ELF4 interacted with eight subunits of this complex (Fig. 6a and Supplementary Data 6).

**Preys linked to RNA splicing and processing in Clusters P and Q.** Next, we found significant enrichment of proteins linked to mRNA splicing and processing in Clusters P and Q (Fig. 5). Cluster P consisted of 16 preys, and Cluster Q consisted of 49 preys, of which 14 and 22, respectively, were linked to RNA splicing and processing (Fig. 6b and Supplementary Data 6). GO-BP analysis showed a significant enrichment of proteins linked to 'mRNA splicing, via spliceosome' ($p$ value $2.6 \times 10^{-7}$ in Cluster P and $1.2 \times 10^{-11}$ in Cluster Q). Cluster P was driven mainly by SP7 (all 16 interactions), GATA1 (15 interactions), and GATA3 (12 interactions) (Fig. 6b). Cluster Q consisted almost totally of SP7 interactions; SP7 interacted with all 49 proteins (Fig. 6b and Supplementary Data 6).

In our dataset, SP7 was the only protein to interact with core spliceosomal components RU17 (SnRP70, U1-70K) and CD2B2 (U5-52K). These newly identified splicing-related interactions indicated that GATA1, GATA3 and especially SP7 are related to splicing and RNA processing. This was also evident in GO-BP enrichment analysis of GATA1 and SP7 interactions using all the

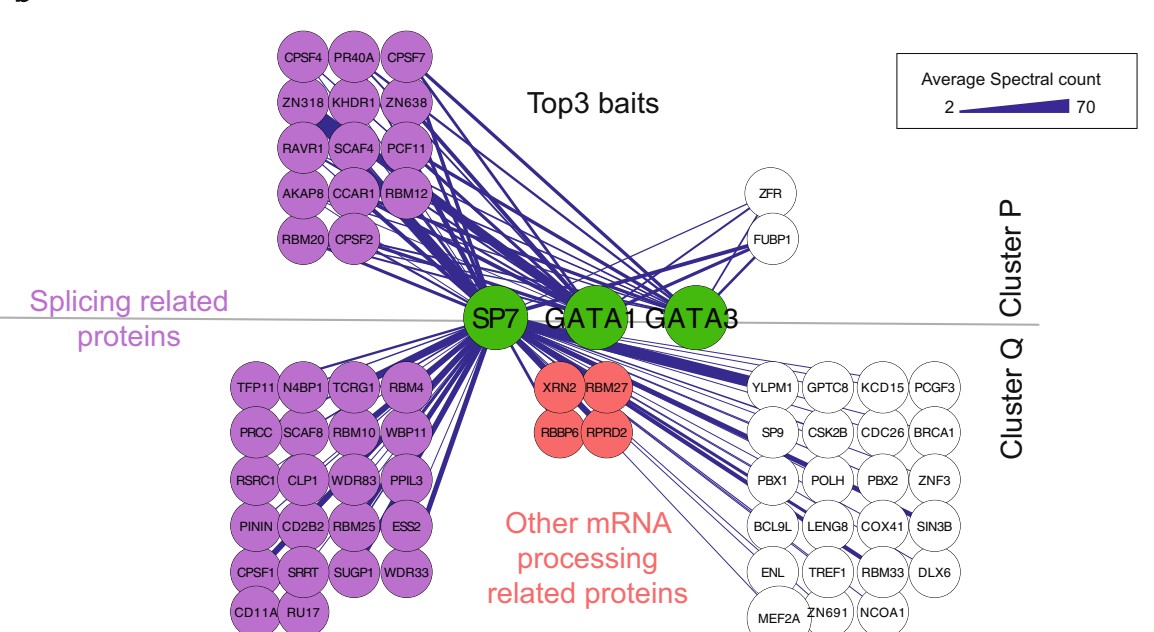

**Fig. 6 The interactions in Clusters K, L, P, and Q of the prey-prey correlation analysis from BioID experiment. a** The ten baits (TYY1, KLF6, HNF4A, KLF8, MYC, NFIX, PAX6, ELF1, ELF2, and ELF4) with the most interactions in Clusters K and L are shown in the top of the figure with green nodes. Most of the preys in Clusters K and L belong to different chromatin-modulating complexes that are highlighted by color coding. **b** Protein–protein interactions in Clusters P and Q. The three baits (SP7, GATA1, GATA3) with the most interactions in Clusters P and Q are shown in the center of the figure with green nodes. Most of the preys in these clusters were linked to mRNA splicing (highlighted in purple). The edge weight displays the average spectral count value detected for the corresponding interaction. Source data of the figure are provided as a Source Data file.

TF interactions from our study as a background; the GO-BP term "mRNA splicing, via spliceosome" was again significantly enriched ($p$ values $1.13 \times 10^{-3}$ for GATA1 and $1.14 \times 10^{-5}$ for SP7; Supplementary Data 4). These new splicing-related interactions for GATAs and SP7 and possible roles in the regulation of splicing are highly intriguing and require further study. In addition to these splicing-related interactions in Clusters P and Q, we found that SP7 and GATA3 interacted with EP300 (p300) along with other members of the p300-CBP-p270-SWI/SNF HAT complex (Supplementary Fig. 4 and Supplementary Data 1b).

## Discussion

Chromatin opening, transcription, RNA splicing, RNA processing, and their regulation are often studied as separate processes. However, our understanding of the simultaneous and cotranscriptional nature of these processes has greatly expanded in recent years[35–39]. In our analyses, TFs were found to interact with proteins involved in chromatin remodeling, transcription, mRNA splicing, and RNA processing, highlighting the cooperative nature and close proximity of these processes. This also showed that TFs are central in regulating these interconnected processes. The most common interaction partners for the TFs studied were histone-modifying enzymes, signifying that histone modification and chromatin accessibility regulation are central to all these transcriptional subprocesses.

The Mediator complex, SAGA complex, and most GTFs are multimeric protein complexes that are needed for Pol-II promoter recognition and transcription initiation[40–42]. In most studied models, PIC assembly starts with the binding of TBP to TATA- or TATA-like core promoters[43]. TBP belongs to two GTF complexes: TFIID and SAGA. It has been indicated that both TFIID and SAGA participate in the transcription of various genes simultaneously[44] but that the regulation of expression might be dominated by either of them[45,46]. Different promoters are alleged to prefer either TFIID or SAGA, and it has been indicated that the activity of SAGA/TATA-like promoters might be more dependent on the presence of transcriptional activators (regulated genes) than the activity of TFIID/TATA promoters (housekeeping genes)[47]. However, this is still controversial, as the depletion of either SAGA or TFIID complex members decreased the transcription of both regulated and housekeeping genes[29,44].

We observed multiple interactions with both TFIID and SAGA components, supporting the theory that both are needed for the transcription of regulated genes. In phase separation condensates, enhancer-bound TFs are physically separated from PIC with multiple cofactor complexes[17–20]. Our data indicated that TFIID and SAGA mainly serve as these cofactors (Supplementary Data 5).

Interestingly, we detected only a few interactions with the Mediator complex (Supplementary Data 5), even though the Mediator is generally thought to mediate the regulatory signals between TFs and Pol-II[48,49]. The mediator complex is reported to interact with multiple TFs, and it is thought to form phase separation condensates with many TFs[20,49]. However, it is not comprehensively known how TFs directly interact with Mediator complex members. Multiple TFs colocalize with Mediator complex members in phase separation complexes in vitro[20]. However, our data indicated that the interactions between TFs and Mediator complex members might be mediated through other proteins, such as histone modifiers. We suggest that for these studied TFs and under the given conditions, the signal is primarily transferred to the Mediator complex and to PIC via other cofactors, such as SAGA, TFIID, or other chromatin remodeling complexes.

Interestingly, we detected a total of 202 bait-bait interactions within the studied TFs (Fig. 4a), most of which (118) were interactions with NFI family members. NFIs are CCAAT-box-binding TFs that have similar DBDs and bind as hetero or homodimers to the same common consensus sequence[32,50,51]. There are four NFI family members (NFIA, NFIB, NFIC, and NFIX) in humans and most vertebrates[52,53]. Originally, NFIs were identified as essential for adenovirus replication[54], but over the years, they have been found to control a variety of genes in cancer and in development[55–60]. For example, NFIs are found to have multiple translocations leading to oncogenic gene fusion proteins in several cancer types[60], and knockout studies of NFIA, NFIB, NFIC, and NFIX have revealed their necessity in lung, central nervous system, brain, tooth, skeletal, and muscle development[55–57,59,61–63].

In our data, SOXs, PAXs, LHXs, KLFs, and TLXs had multiple interactions with NFIs (Supplementary Data 1b). These interactions indicated that NFIs take part in many cellular processes with other TFs. NFI family members have been found to interact with some individual TFs, such as with a few SOX family proteins[64,65]. However, TF-NFI interactions on this scale have not been reported before. Given the important role of NFIs in the regulation of developmental processes and their impact on cancer development, the high number of TF-NFI interactions might indicate that the activity of NFIs is generally regulated by other TFs, or vice versa. To test this hypothesis, we generated several luciferase reporter assays specific for selected NFIA-interacting TFs and discovered that RNAi silencing resulted in altered corresponding TF activity (Fig. 4d). This supports the theory that NFIs' interactions with other TFs have extensive and not well-characterized roles in the regulation of other TF activities in gene expression regulation.

Of note, we recently studied immunodeficiency patients with Y200X variant of IKZF2 transcription factor. This variant displayed clear interaction reduction with NFIA, NFIB, and NFIC. Furthermore, RNA-seq analysis of the patients and healthy controls T-cells indicated immune dysregulation of several major immune activation pathways in the patients[66].

In our analysis, proteins involved in RNA splicing were enriched in the interactomes of SP7, GATA1, and GATA3. In addition, we found that SP7 and GATA3 interacted with EP300 (p300) along with other members of the p300-CBP-p270-SWI/SNF HAT complex (Supplementary Fig. 4 and Supplementary Data 1b).

TFs are known to affect RNA splicing in three ways: they can bind to RNA to recruit coregulators that also take part in splicing, block the associations of splicing factors with mRNA, and/or influence transcription elongation rates, which are known to impact splicing by skipping weak 3' splice sites at a high rate[67]. One way to alter the elongation rate is through TF-mediated recruitment of EP300, which induces the histone acetylation of nearby promoters, increases the elongation rate and promotes exon skipping[68]. Therefore, the interaction of SP7 and GATA3 with EP300 suggests that they may be connected to the p300 chromatin remodeling complex and, thus, to the regulation of the elongation rate.

Some TFs, such as steroid hormone receptors, nuclear receptors (NRs) and certain non-NR TFs, are known to regulate mRNA splicing by recruiting splicing-linked coregulators[67,69]. One of these coregulators is RBM14 (also known as CoAA), which is an NCOA6 (also known as TRBP)-binding protein[70]. We found that, along with other splicing-related proteins, NCOA6 interacted with SP7 (Supplementary Data 1b). This might indicate that SP7 recruits similar splicing regulation-related coregulators.

SP7 interactions with core spliceosomal components and all other splicing-related components suggested that SP7 has a largely unstudied role in recruiting the splicing machinery to the nascent pre-mRNA – a role that needs to be further studied. Similar to some other C2H2 zinc finger TFs, such as CTCF, VEZF1, MAZ, and WT1, which are known to regulate mRNA splicing[67], SP7 might participate in pre-mRNA splicing.

As expected, multiple TFs interacted with chromatin-modulating proteins. We found several TFs to interact, e.g., with INO80, NSL, SAGA, and NuA4/Tip60 HAT complexes (Fig. 6a and Supplementary Data 1b). TF interactions with these complexes will be discussed in more detail.

The INO80 is an ATP-dependent chromatin remodeling complex that activates transcription[34], regulates genomic stability through DNA repair[71], contributes to DNA replication[72], and by shifting nucleosomes, remodels chromatin[73,74]. Predictably, TYY1, which is known to be part of the INO80 complex, had multiple interactions with INO80 complex members in our analyses. More interestingly, we found that ELF4 also had multiple interactions with INO80 members. To our knowledge, ELFs have not been previously linked to INO80 signaling or TYY1, and this should be studied further.

Next, we found that various TFs, such as TYY1, ELF1, ELF2, ELF4, HNF4a, and MYC, interacted with members of the NSL histone acetyltransferase complex. Interactions between TYY1 and three NSL complex subunits (MCRS1, HCFC1, WDR5) have been reported (PINA2[75]), but the role of TYY1 in NSL regulation is still unknown. Moreover, the roles of ELF1, ELF2, and ELF4 in the NSL complex (as in the INO80 complex) and their interactions with WDR5 are not well understood and require further study.

As mentioned earlier, the SAGA complex is a multimodule complex that has an important role in Pol-II recruitment for all expressed genes[76]. In addition, SAGA participates in mRNA synthesis and export, maintenance of DNA integrity, and histone modifications, such as histone acetylation, succinylation, and ubiquitylation[76–82]. In our data, multiple interactions were detected between SAGA complex members and the studied TFs (e.g., MYC, KLF6, KL8, HNF4a; Tables S1B and S5). MYC connections to SAGA are known[83], and KLF6 interactions with one of the subunits, TAF9, have been reported in the PINA2 database. In addition, KLF6 is known to interact with HDAC3 in preadipocyte differentiation[84]. However, we found no other connections between KLF6 and the SAGA complex or histone modification. Our data indicated that KLF6 connections to SAGA are bona fide and should be studied further.

Finally, we found MYC, HNF4a, KLF6, KLF8, and ELF4 to interact with the NuA4/Tip60 HAT complex (Tables S1B and S7), which plays essential roles in cell cycle control, transcription and DNA repair and acts in the N-terminal acetylation of histones H4 and H2A[85]. The NuA4/Tip60 HAT complex, along with other HAT complexes, is known to participate in MYC signaling[86]. Accordingly, we found 14 interactions between MYC and the NuA4/Tip60 HAT complex (Tables S1B and S7). Similar to MYC, HNF4a's association with the NuA4/Tip60 HAT complex was reported in a previous study[87]. Interestingly, we also found a strong connection (eight to nine interactions) between KLF6, KLF8, or ELF4 and the NuA4/Tip60 HAT complex. However, as mentioned earlier with regard to the SAGA complex, little is known about the link between KLF6 and histone modification. Therefore, the role of KLF6 in HAT complexes remains largely unstudied and requires further investigation.

Taken together, TYY1, ELF4, ELF1, ELF2 (Cluster K) and MYC, KLF6, KLF8, and HNF4a (Cluster L) had several interactions with chromatin remodeling complexes. Some research has been conducted on the contributions of TYY1, MYC, and HNF4a

to histone modification and chromatin remodeling, but the roles of ELFs and KLFs in chromatin remodeling remain largely unexplored.

Interestingly, even though chromatin remodeling and histone modifications are known to be important for almost all TF signaling and most of the studied TFs interact with proteins that mediate these processes, only a fraction of TFs seemed to interact with almost complete histone-modifying or chromatin remodeling complexes. These interactions include TYY1 and ELF1 interactions with the INO80 complex; TYY1 and MYC interactions with the NSL complex; MYC and KLF6 interactions with the SAGA complex; and MYC, HNF4a, KLF6, KLF8, and ELF4 interactions with the NuA4/Tip60 HAT complex. This observation suggests that these TFs act in close relation to these complexes and take part in these complexes at least under certain conditions.

While TF binding to DNA is well-studied, there is still a lack of comprehensive systems-level understanding of human TF protein interactions. The protein interactions of other large human protein families, such as kinases and phosphatases, have been studied at the systems level[16,21,22,88], but TF protein interactions remain globally unknown. This study provides the most comprehensive systems-level analysis of human TFs, identifying the largest reported cohort of TF PPIs and serves as a rich resource for further research and development of pharmaceutical treatment for TF-related diseases. This work also allows the profiling of TF protein interactomes in the context of more than 100 TF interactomes. Moreover, this is the first large-scale study to identify the dynamic PPIs of TFs using transient and proximal interactions catching BioID method. Finally, as abnormalities in TF signaling often lead to severe pathological conditions[89–91] and TFs function as downstream players of multiple signaling cascades[3], the identification of TF PPIs makes a crucial contribution to pharmacological targeting of TF-related diseases. Although our study represents currently the most comprehensive interactome analysis of the human transcription factors, we acknowledge some limitations of this current study. These include that the cellular context might be ectopic to some TFs, as well as that some TFs require a certain cellular state (such as cell cycle stage) for interactions and activity. From the established and used cell lines, HEK293 express a wide range of different transcript, totaling ~13k genes. In interaction proteomics HEK293 cells have been widely used as a "gold standard" for studying protein–protein interactions (PPIs)[27,92,93]. The proteome functional classification of HEK293 mimicks UniProt and most closely resembled the distribution observed for preys[92]. Previous study has shown that even with the lentiviral infection of 293 cells, overexpression has little effect on identification of true interacting partners after statistical filtering[94].

Additionally, several of the TFs also can display cell cycle-dependent transcriptional regulation and hence interactions. These interactions should also be captured with our approach as during the induction the cells display heterogenous cell cycle stages and furthermore the labeling time used in the BioID approach covers one full cell cycle.

In summary, our study provides a comprehensive and complimentary overview of on the physical and functional interactions of the transcription factors. Our findings pave the way for further studies using different cell lines, organoids, or even transgenic animals.

## Methods
**Cell lines used**. HEK293 cells have been widely used for the study of protein–protein interaction (PPI)[27,92,93]. The proteome functional classification of HEK293 mimicked UniProt and most closely resembled the distribution observed for preys[92]. Previous study has shown that even with the lentiviral infection of 293

cells, overexpression has little effect on the identification of true interacting partners after filtering[94].

In this work, with Flp-In™ 293 T-Rex (Invitrogen, Cat# R78007), allowing to generate isogenic and inducible stable cell clones with only a single copy of a transgene in their genome, provides a convenient method to study PPIs[95]. siRNA silencing, luciferase experiments, and CoIP-DB analysis were performed in the HEK293 cell line (American Type Culture Collection, Manassas, VA, Cat# ATCC CRL-1573). All cells were cultured in low glucose tetracycline-free DMEM (Sigma Aldrich) supplemented with 10% FBS and 100 µg/ml penicillin/streptomycin (Life Technologies) at 37 °C with 5% $CO_2$.

**Generation of TF expression constructs and stable, inducible cell lines**. A total of 109 TFs from different TF families were selected for this study. Using Gateway® cloning, TF coding sequences without stop codons were obtained from ORF libraries and commercially cloned into pDONR221 entry vectors (GenScript). To generate tetracycline-inducible stable cell lines, constructs were cloned into N-terminal pTO_HA_StrepIII_BirA-N_GW_FRT, pTO_HA_StrepIII-N_GW_FRT, or MAC-tagged vectors and introduced into Flp-In™ T-REx™ 293 cells (Life Technologies, Carlsbad, CA) to generate stable, isogenic, and inducible cell lines as described by Liu et al.[15].

**Affinity purification and mass spectrometry analysis**. Approximately $1 \times 10^8$ Flp-In™ T-REx™ 293 cells stably expressing human TFs were induced with 2 µg/ml tetracycline (AP-MS and BioID) and 50 µM biotin (BioID) for 24 h. The cells were pelleted using centrifugation, snap frozen in liquid nitrogen, and stored at −80 °C. The samples were then suspended in 3 ml of lysis buffer (50 mM HEPES pH 8.0, 5 mM EDTA, 150 mM NaCl, 50 mM NaF, 0.5% NP40, 1.5 mM $Na_3VO_4$, 1 mM PMSF, 1x protease inhibitor cocktail, Sigma) on ice.

BioID lysis buffer was completed with 0.1% SDS and 80 U/ml benzonase nuclease (Santa Cruz Biotechnology, Dallas, TX), and lysis was followed by incubation on ice for 15 min and three cycles of sonication (3 min) and incubation (5 min) on ice.

All samples were then purified by centrifugation, and the supernatants were poured into microspin columns (Bio-Rad, USA) that were preloaded with 200 µl of Strep-Tactin beads (IBA GmbH) and allowed to drain under gravity. The beads were washed 3x with 1 ml lysis buffer and then 4x with 1 ml lysis buffer without detergents and inhibitors (wash buffer). The purified proteins were eluted from the beads with 600 µl of wash buffer containing 0.5 mM biotin. To reduce and alkylate the cysteine bonds, the proteins were treated with a final concentration of 5 mM TCEP (tris(2-carboxyethyl) phosphine) and 10 mM iodoacetamide, respectively. Finally, the proteins were digested into tryptic peptides by incubation with 1 µg sequencing grade trypsin (Promega) overnight at 37 °C. The digested peptides were purified using C-18 microspin columns (The Nest Group Inc.) as instructed by the manufacturer. For the mass spectrometry analysis, the vacuum-dried samples were dissolved in buffer A (1% acetonitrile and 0.1% trifluoroacetic acid in MS grade water).

The peptides were analyzed on an EASY-nLC II system connected to an Orbitrap Elite ETD hybrid mass spectrometer (Thermo Fisher Scientific, Waltham, MA) using Thermo Scientific™ Xcalibur™ Software (version 2.7.0). The digested peptides were first guided into a precolumn (C18-packing; EASY-Column™ 2 cm × 100 µm, 5 µm, 120 Å, Thermo Fisher Scientific) and then into an analytical column (C18-packing; EASY-Column™ 10 cm × 75 µm, 3 µm, 120 Å, Thermo Fisher Scientific). The separation was completed with a 60-min linear gradient from 5 to 35% buffer B (98% acetonitrile, 0.1% formic acid, and 0.01% trifluoroacetic acid in MS grade water) at a stable flow rate of 300 nl/min. Data-dependent acquisition analysis was performed as follows: after one high-resolution (60,000) FTMS full scan (*m/z* 300–1700), the top 20 CID-MS2 scans in the ion trap were performed (energy 35). The highest fill time was 200 ms for FTMS (full AGC target 1,000,000) and 200 ms for the ion trap (MSn AGC target of 50,000). Only precursor ions with ion counts higher than 500 were chosen for MSn. The preview mode was applied for the FTMS scan to achieve a high resolution. Finally, dynamic exclusion was enabled, and the settings were set as follows: repeat count: 1, repeat duration: 30.00, exclusion list size: 500, exclusion duration 30.00, exclusion mass width relative to low (ppm): 5.000 and exclusion mass width relative to high (ppm): 5.000.

**Protein identification**. The proteins were identified using the SEQUEST search engine in Proteome Discoverer™ software (version 1.4, Thermo Scientific). The raw data were analyzed against the reviewed human proteins from the UniProt database (release 2018_01; 20,192 entries)[96]. The FASTA library was complemented with BSA, tag sequences, trypsin, and GFP. Biotinylation (+226.078 Da) of lysine residues and oxidation (+15.994491 Da) of methionine were used as dynamic modifications. In addition, cysteine residue carbamidomethylation (+57.021464 Da) was used as static modification. A maximum of two missed cleavages and 15 ppm monoisotopic mass error were allowed. The peptide false discovery rate (FDR) was set to <0.05. The precursor mass tolerance was set to 15 ppm, and the fragment mass tolerance was set to 0.05 Da.

The identified proteins were filtered using SAINT software tools (SAINTexpress version 3.1.0)[97] with a SAINT score cutoff of 0.74. All the TFs were analyzed in two or four replicates. In addition, we added control data from the CRAPome database

(version 2.0)[23] from 716 experiments to further filter possible contaminants from the list. In the BioID data, only proteins with a lower frequency than 50% (358/716) in the CRAPome were allowed. Additionally, prey with a CRAPome frequency of 25–50% (179–358/716) and with a higher CRAPome average spectral count compared to our average spectral count were removed.

From AP-MS data, in addition to the SAINT cutoff, prey that was present with a frequency higher than 50% in the CRAPome (358/716) were required to have a threefold higher average spectral count than the average spectral count in the CRAPome database.

TFs are known to have variable expression levels and patterns, and some of them are present in cells at extremely low copy numbers[98]. To efficiently filter the real interactions, we used 44 and 75 similarly tagged and analyzed GFP control runs for the BioID analysis and AP-MS analysis, respectively. From these, 16 and 18, respectively, had a nuclear localization signal (NLS) to efficiently filter out nonspecific nuclear interactions. All GFP controls were used as negative controls in SAINT analysis, where the large nuclear dataset further facilitated the frequency-based deletion of contaminating proteins. The Cytoscape software platform (version 3.8.2) was used to visualize the high-confidence TF PPIs[99].

**Data analysis**. The subcellular localizations of interacting proteins were obtained from the Cell Atlas[12]. Enriched biological process Gene Ontology terms for all PPIs were obtained from DAVID Bioinformatics Resources[100]. We also used DAVID to study the enrichment of separate TF interactomes against all the PPIs identified in our study. All the terms with the corresponding p values and FDR are reported in Supplementary Data 4.

Correlation for biological and technical replicates was analyzed using spectral count values of either biological or technical replicates. Pearson's correlation coefficient values were calculated for each pair of replicates with the pearsonr method from the python scipy.stats package (SciPy, version 1.71). Plots for the results were generated with lmplot method of the python seaborn package (version 0.11.2).

Identified PPIs were compared to PPIs from public interaction databases such as PINA2 (version 2.0)[24], STRING (version 11)[101], IntAct[102] and BioGRID (version 4.4)[103], and PPIs from several medium- to large-scale interactome studies such as Lambert et al.[25], Malovannaya et al.[26], and Huttlin et al.[27]. CORUM database[104] was used for protein complex analysis.

Hierarchical clustering of baits (studied TFs) by their prey (interacting proteins) was performed using ProHits-viz. with default settings[33]. Comparisons of two cluster dendrograms were performed using the *dendexted* R package (version 1.14.0, https://www.datanovia.com/en/lessons/comparing-cluster-dendrograms-in-r/). The full amino acid sequences of the studied TFs were downloaded from UniProt[96]. The DNA-binding motifs of the studied TFs were mainly extracted from the JASPAR database (7th release)[31]. Motifs not found in JASPAR were extracted from the HT-SELEX and ENCODE databases[105,106]. All extracted DNA-binding motifs were aligned using the matrix-clustering tool RSAT[28]. Finally, the prey-prey correlation analysis of the BioID data was performed using ProHits-viz.'s correlation tool (https://prohits-viz.lunenfeld.ca/Correlation/), where Pearson correlation and hierarchical clustering with Euclidean distance metric was used[33]. Filtered SAINT-interactions were used as the input. Apart from default settings, the score column was set to SAINTScore, and cutoff values for filtering were removed as filtered interaction data were used as input.

TF activity was assessed by luciferase assays in three replicates (Figs. 4D and S4). Firefly luciferase signals were normalized with Renilla luciferase signals, and Student's *t* test was used to detect the significance of the changes. Asterisks in the figs. (Figs. 4D and S4) indicate the following cutoffs: ***$p < 0.001$, **$p < 0.01$, *$p < 0.05$.

**Co-immunoprecipitation**. HEK293 cells were seeded into six-well plates at a density of $5 \times 10^5$ per well and incubated overnight. The cells were cotransfected with Strep-HA-tagged (500 ng) NFIA, NFIB or NFIC and one V5-tagged (500 ng) prey construct using Fugene 6 transfection reagent (Promega). Twenty-four hours after transfection, cells were harvested, washed with ice-cold PBS and lysed with 1 ml HENN lysis buffer per well (50 mM HEPES pH 8.0, 5 mM EDTA, 150 mM NaCl, 50 mM NaF, 0.5% IGEPAL, 1 mM DTT, 1 mM PMSF, 1.5 mM Na3VO4, 1× protease inhibitor cocktail) on ice. The cell lysate was collected, and protein extracts were collected following centrifugation at 16,000 × g for 20 min at 4 °C. Meanwhile, 40 µl of Strep-Tactin® Sepharose® resin (50% suspension, IBA Life-sciences GmbH) was washed in a microcentrifuge tube twice with 200 µl lysis buffer (4000 × g, 1 min, 4 °C). The clear cell lysate was added to the prewashed Strep-Tactin beads and incubated for 1 h on a rotation wheel at 4 °C. After incubation, the beads were collected by centrifugation and washed three times with 1 ml lysis buffer (4000 × g, 30 sec, 4 °C). After the last wash step, 50 µl of 2× Laemmli sample buffer (Bio–Rad, 1610737) was added onto the beads, and the bound complexes were eluted by boiling for 5 min at 95 °C, followed by dot blot analysis.

**Dot blot**. The Bio-Dot® Microfiltration System (Bio–Rad, 1703938) was assembled according to the manufacturer's instructions. The nitrocellulose membrane was prewashed with TBS to hydrate the membrane. Ten microliters of sample was spotted onto the nitrocellulose membrane in the center of the well and drained

under vacuum pressure. The membrane was blocked with 5% skim milk in TBS-T (Tris-buffered saline with 0.05% Tween 20) for 1 h at room temperature (RT) with gentle shaking. Then, the membranes were incubated with the respective primary antibodies (mouse anti-V5, ThermoFisher, Cat# R960-25, with a 1:5000 dilution in blocking solution and mouse anti-HA, BioLegend, PRC-101C, with a 1:2000 dilution in blocking solution) overnight at 4 °C with gentle shaking. After three 10 min washes with TBS-T, the membranes were incubated with secondary antibodies conjugated with HRP (Goat anti-mouse IgG H&L (HRP), Abcam, Cat# 97023, with a 1:2000 dilution in blocking solution and Goat anti-rabbit IgG H&L (HRP), Abcam, Cat# ab205718, 1:2000 dilution in blocking solution) for 60 min at RT with gentle shaking. The membrane was washed three times for 10 min with TBS-T followed by one additional wash with TBS on a shaker. For visualization, Amersham™ ECL™ Prime (Cytiva) solution was added to the membrane and incubated for 5 min prior to imaging the blot using iBright Imaging Systems (Thermo Fisher). The same membrane was then stripped by incubating with Restore Plus Stripping buffer (Thermo Fisher) for 15 mins and was reblocked with the blocking solution for 60 min at RT with gentle shaking. The membrane was then incubated with the other primary antibody (mouse anti-HA with a 1:2000 dilution in blocking solution) overnight at 4 °C with gentle shaking, and the process was carried out as before.

**NFIA silencing and reporter gene assays**. HEK293 cells were cultured in 96-well plates (7000 cells/well) for 24 hours. This was followed by NFIA siRNA (Dharmacon J-008661-06) transfection at a final concentration of 100 nM using Dharmafect transfection reagent (0.35 μl/well). After 24 h of siRNA silencing, the culture medium was replaced with fresh medium, and cells were transfected with 50 ng of the selected TF or empty vector (pTO-SH-GW-FRT) along with 47.5 ng of reporter construct. The reporter constructs contained 6-8x TF binding sites (TFBSs[31], minimal promoter and firefly luciferase reporter. Only the constructs displaying induction after introduction of the corresponding TF were chosen for further analysis. These included reporters for KLF4 (both activating: [TFBS: 6x GGGTGTGG] and repressive: [TFBS: 8x TAAAGGAAGG]), SOX2 (TFBS: 6x CTTTGTT), PAX6 (TFBS: 6x TTCACGCTTGAGTT) and HME1 (TFBS: 8x AAGTAGTGCCC).

In addition, cells were transfected with 2.5 ng of Renilla-luciferase construct. After 24 hours, the cells were collected, and the firefly luciferase and Renilla luciferase signals were detected using a Dual-GLO® luciferase assay system (Promega). Firefly luciferase signals were normalized to Renilla luciferase signals, and the analysis was performed in three replicates. NFIA silencing was confirmed 48 hours after siRNA transfection by western blotting using a specific antibody against NFIA (Abcam, ab228897, 1:1000) and anti-Rabbit antibody (Dako, Cat# P0448, 1:1500).

**Reporting summary**. Further information on research design is available in the Nature Research Reporting Summary linked to this article.

## Data availability

The source data of the figures are provided with this paper as a separate Excel sheet. The MS peptide raw data from the MS runs have been deposited in the Massive database (http://massive.ucsd.edu/ProteoSAFe/status.jsp?task=0bfbe238f2ab4bd1a12fec75e4f6c67e) under accession number MSV000086891, and the protein interactions from this publication have been submitted to the IMEx (http://www.imexconsortium.org) consortium through IntAct[102] and assigned the identifier IM-28767. Filtered protein–protein interactions are also available in Supplementary Data 1. Following databases were used in data analysis: Uniprot (release 2018_01; 20,192 entries, https://www.uniprot.org), CRAPome 2.0 (http://www.crapome.org/), PINA Interaction network analysis tool (version 2.0, http://cbg.garvan.unsw.edu.au/pina/), STRING: functional protein association network database, version 11, (https://string-db.org), IntAct Molecular Interaction Database (https://www.ebi.ac.uk/intact/home, downloaded 30.4.2021), BrioGRID, Database of Protein, Genetic and Chemical Interactions (version 4.4, https://thebiogrid.org), Mammalian protein complex resource: CORUM Institute of Bioinformatics and Systems Biology (http://mips.helmholtz-muenchen.de/corum/), JASPAR database (7th release, https://jaspar.genereg.net), HT-SELEX database (https://ccg.epfl.ch/htpselex/), and ENCODE database, (https://maayanlab.cloud/Harmonizome/dataset/ENCODE+Transcription+Factor+Targets). Source data are provided with this paper.

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

## Acknowledgements
We would like to thank the Orion Research Foundation sr, Finnish Cultural Foundation (HG), Academy of Finland (288475 and 294173), Sigrid Jusélius Foundation, Biocentrum Helsinki, University of Helsinki Three-year Research Grant, Biocentrum Finland, HiLIFE, Instrumentarium Research Foundation (MV), Jane and Aatos Erkko Foundation and Finnish Cancer Foundation (GW). We would also like to thank Petri Auvinen and Juha Kere for the critical reading of the manuscript.

## Author contributions
H.G., M.K., and M.V. designed the study. H.G. and M.K. with the help of L.Y. generated the cell lines, and H.G. and M.K. performed the mass spectrometry analyses. H.G., G.W., Z.T., and Q.Z. performed RNAi analyses for the NFIs. H.G. performed the data filtering. H.G., K.S., and M.V. did the data analysis. L.X. and M.K. performed the dot plot analysis and analyzed the results. H.G. and M.V. prepared the figures and wrote the manuscript. All authors reviewed the manuscript carefully.

## Competing interests
The authors declare no competing interests.
