## [Peer Review File · Nature Communications]

Reviewers' Comments:

Reviewer #1:

Remarks to the Author:

In their paper, Goos et al. describe their efforts to map protein interactions for 110 human transcription factors using both AP-MS and BioID. The result of this effort is a sizeable interaction network, most of which has not been previously described, and which provides numerous insights into the biology of individual transcription factors and transcription complex organization more generally. The authors select a couple of specific examples for targeted follow-up, looking for effects of NFIA silencing on activity of several transcription factors to which it binds. They also used ChIP-seq to show that NFIA silencing impacts SOX2 binding to chromatin. Finally, they mine their network for additional biological insights using various bioinformatic approaches, several of which they discuss in some detail. While I expect this work will eventually be of broad interest to the scientific community, I have some technical concerns and questions that need to be addressed before this paper will be ready for publication.

MAJOR COMMENTS

1. As the authors point out, there are ~1600 human transcription factors. Since just a fraction of these (110) were characterized in this study, how these were selected and how representative they are of the larger group will have important implications for how this network can be used. How did the authors decide which transcription factors to target in this study?
2. Close examination of Table S1B-C raises some questions about how the data were filtered to identify significant interactions. First, in both BioID and AP-MS datasets, the target baits are themselves reported as interacting proteins. For example, at the top of Table S1C we find the protein BRAC listed in the "Prey" column for the BRAC pulldown. It's also listed as a prey when it is the target of BioID analysis as well (Table S1B). This is true for most or all baits in both AP-MS and BioID datasets. That these bait proteins are detected – and strongly enriched – is to be expected, since they are being specifically targeted for purification. However, labeling these proteins as "preys" in their own IP's and including these observations in final interaction lists and total interaction counts for the dataset as the authors have done is misleading. People accessing these data in the future may erroneously take these entries to imply homo-dimerization, especially once these data are incorporated into databases like BioGRID and IntACT. These entries should be removed from the supplementary tables and the total interaction counts throughout the text adjusted accordingly.
3. A second concern emerging from examination of Table S1B is that at least one obvious contaminant protein seems to be detected repeatedly as an interacting partner. Bovine serum albumin (BSA) from cell culture media is identified as an interacting partner via BioID for 11 different transcription factors, representing 10% of the dataset: ELK4, FOS, FOXQ1, GATA3, GLI2, HME1, LHX6, MYC, MYOD1, SMAD5, and STAT3. These are only detected in the BioID dataset. I didn't find BSA listed at all in the AP-MS interaction list. This raises a few points. First, the authors should filter out these interactions since they're clearly artifacts. More generally, if BSA passed as an interacting protein in 10% of their BioID experiments, this raises questions about whether their BioID platform is particularly susceptible to interference from background proteins and to what extent other contaminating proteins remain in their filtered interaction lists. Are the authors sure they've sufficiently dealt with these artifacts?
4. Starting on line 244, the authors highlight a cluster of actin/myosin-related prey proteins in the BioID dataset that cluster together and link to several transcription factors. The authors interpret these interactions as evidence of a link between these transcription factors (especially FOS and STAT1) and actin/myosin signaling in the nucleus. Perhaps this is the case - there is evidence that some of these proteins (ACTB, MYO1C, etc) do reside in the nucleus where they perform functions related to chromatin remodeling and transcription regulation. However, a skeptic might wonder whether most of these actin/myosin-related proteins are instead artifacts for several reasons. First, though a few of these proteins are known to reside partially in the nucleus and participate in transcription-related processes, many are not. (Examples include MYO1B, MYO1D, MYL6B, MYH9, MYH10, MYO5A, etc. according to Uniprot and Human Protein Atlas as listed in Table S2 of this

paper.) Second, actin, myosin, and related proteins are themselves abundant and are often detected as background in these kinds of experiments. Third, there may in fact be elevated background in the BioID experiments targeting baits associated with Cluster 2. As I mentioned above, the contaminant BSA is present at elevated levels and flagged as an interacting partner in BioID experiments associated with 11 baits, reflecting anomalous background. Nine of these baits are linked to this actin/myosin cluster (Cluster 2 - see Table S7). Fourth, relatively few of the preys in this cluster appear to be nuclear, while several (PGRC1, TIM50, RAB8A, ATPA, ATPB, AT1A1, ADT2, ENPL, CALX, etc) localize to membranes, mitochondria, ER, and other alternate compartments and would seem to be contaminants (Table S7). Finally, while the BioID data for FOS and STAT1 reveal many proximal actin/myosin proteins, none of these interactions appear in the AP-MS data. While this could mean that these interactions are too transient or weak to survive AP-MS purification, it could also mean that these are artifacts specific to these BioID samples, much like BSA and the membrane/mitochondrial/ER preys detected alongside them. If the authors want to highlight this cluster, they will need to make a stronger case that these observations represent biology and should provide additional independent experimental confirmation to show that these links between FOS/STAT1 and actin/myosin proteins are real. Alternatively, they could cut this example and focus instead on other clusters where the data are more convincing.

5. I also note that the authors report that the mitochondrial carboxylase ACACB interacts with about 22 different transcription factors. This is also likely an artifact, as this protein is endogenously biotinylated according to Uniprot and Lee et al. (2008) *Proteins* 72:613-24. I should also point out that all 22 of these IP's with excessive ACACB levels are associated with Cluster 2, seemingly linking that cluster to another experimental artifact. Endogenous biotinylation may also explain the presence of some other likely artifacts reported as interactions in the BioID data, including mitochondrial carboxylases MCCA, MCCB, PCCA, and PCCB that are covalently modified with biotin (e.g. Samavarchi-Tehrani et al. (2020) *Mol Cell Proteom* 19:757-773).

6. In various places in the text the authors contrast BioID and AP-MS, saying for example that BioID "... was used to detect also transient and proximal interactions of the TFs." (Line 96). And "The BioID method has been suggested to be efficient for studying transient interactions..." (Line 111). These statements are true, though the authors should point out explicitly that as long as they're close together, a prey protein doesn't actually need to physically associate with a given bait to be labeled via BioID. And given the confined nature of chromatin, these TF's might be unusually prone to labeling other proteins that are in close proximity but don't actually interact.

MINOR COMMENTS

1. I'm curious about the authors' observation that relatively few of the interactions they saw had been previously reported (Figure 1C). Is this fraction higher for really well characterized transcription factors (e.g. P53, MYC, etc) versus other less thoroughly characterized transcription factors?

2. Figure 1D: It looks like about 20 baits are missing from this plot. Why aren't all 110 shown? If the authors can't fit them all in the plot, then they should at least indicate in the figure legend that only a subset of baits are shown. Also, there seems to be a typo - one bait is simply labeled as "T".

3. Line 140: I'm not sure it's fair to say co-IP isn't sensitive enough. The issue is that AP-MS and BioID are measuring two phenomena that are related but distinct. One measures physical binding while the other measures proximity.

4. Figure 2: I'm having trouble distinguishing the blue and black edges. I'd change the color scheme for improved contrast. The same is true for Figure 4A.

5. Figure 4C. It seems that HME1 is an alternative name for EN1 (both labels are given in Figure 2). Why is there a row labeled "HME1" in the PAX6 plot? This looks like a mislabeling. Also, the authors should use consistent terminology in the panel labeled EN1 (Lower lefthand corner of Figure 4C) - right now the same protein is labeled EN1 at the top of the plot and HME1 below the bar graph.

6. In the Methods section titled "Protein identification" the authors write "FASTA library was complemented with BSA, tag sequences, trypsin, biotin, and GFP." Since biotin is a small molecule, perhaps the authors mean they included streptavidin (in place of strep-tactin, whose sequence is presumably proprietary)?

7. In the "Protein identification" section the authors also write that "oxidation ... of methionine or N-terminus were used as dynamic modification". Oxidation of the N-terminus doesn't make chemical sense in this context. Do the authors mean acetylation of the N-terminus? Also, what was the product ion mass error?

8. Instrumentation Methods (lines 522 – 533): What dynamic exclusion settings were used on the instrument?

9. In line 544 the authors write that "All the TFs were analyzed in two or four replicates". What determined whether each TF was analyzed two versus four times? Were these biological or technical replicates? Were replicates performed separately or side-by-side?

10. Line 542: the authors say they filtered peptides to an FDR <5%. What was the resulting protein-level FDR? And what additional steps (if any) were taken to control the protein-level FDR?

11. Line 546. The authors write "...we used 44 and 75 similarly tagged and analysed GFP control runs for the BioID analysis and AP-MS analysis, respectively. We also included GFP's with a nuclear localization signal..." Are the authors saying that those 44 and 75 GFP control runs in BioID and AP-MS were performed using GFP with a nuclear localization signal? Or were the analyses of GFP's with a nuclear localization signal separate from those listed in the preceding sentence?

Reviewer #2:

Remarks to the Author:

The manuscript by Göös et al. with title of "Human transcription factor protein interaction networks" presented a comprehensive interactome profiling of 110 human transcription factors using both AP-MS and BioID approaches. As the authors well summarized that transcription factors represent 6-9% of human proteome and play crucial roles in gene transcription related mechanisms. A systematic profiling of such type of protein machinery is desirable and should provide a valuable resource to related biological fields. The combination of traditional AP-MS analysis in cell lysate and proximity labeling-proteomic analysis in living cells should provide complimentary information for better viewing their molecular functionality. However, current manuscript didn't present sufficient information to support the above goals and therefore needs to be significantly improved before further consideration.

Major comments:

1. Interactome profiling of transcription factors has been a popular topic in the past decade especially by using standard AP-MS approach. However, the authors failed to systematically review and summary these early efforts and clearly present the novelty of their own study. For example, yeast chromatin-associated interactome by Figeys and co-workers in 2010 (*Mol. Syst. Biol.*, 6, 448) and large-scale endogenous IP-MS analysis of transcriptional proteins by Qin and co-workers in 2011 (*Cell*, 145, 787). In addition, continuous global-scale AP-MS analysis of more than 10,000 human proteins by the BioPlex project leading by Gygi and co-workers should also cover a significant fraction of transcription factors (BioRxiv, <https://doi.org/10.1101/2020.01.19.905109>). Except for expanding early efforts for profiling transcription factor interactome, the authors should deliver more clear biological rationale for such type of study and major biological conclusion.

2. Along this line, the authors didn't explain why the 110 transcription factors were selected for this specific study. Are there any previous AP-MS studies which have studied some of them? The authors should perform a related bioinformatic analysis and provide extended discussion about the rationale for studying this limited fraction of transcription factors.

3. As a valuable resource, the data quality is critically important especially for BioID type of data which represent the proximal proteome and tends to generate significant amount of nonspecific labeling. The authors only presented very general overview of the design of the study. Current data presentation failed to convince the audience. The authors have to significantly expend their data analysis dedicated for data quality part. Specifically, for example, the authors mentioned two or four replicates were performed. Why it is not consistent for such a critical experimental design? How the data reproducibility was evaluated and controlled? The authors used 44 and 75 GFP control runs for the BioID analysis and AP-MS analysis, respectively. How did these important control data use for filtering out nonspecific adsorption and labeling? For all the figures in this study, the authors only presented the interaction network data. It's hard to conclude the reliability of the highlighted interaction hubs – associated peptide counts etc. should be properly presented.

4. In Fig. 1C, the authors highlighted much higher number of identified PPI by the BioID approach. As stated before and the cited papers by the authors, it's not proper to define the identified proteins as PPI, but only proximal proteins. Along this line, the authors should extend their discussion about how the nonspecific labeling was properly filtered? Any more accurate quantification was used, which has becoming the standard approach in the proximity proteomics field.

5. Line 104, the authors highlighted >75% TF interactome were previously unreported. The authors should specify the resource for such a comparison. In the section of "Comparison to other studies", the authors well compared with Li et al.'s work for studying 59 TF interactome. However, only 6% of PPIs identified by BioID and 26% of PPIs by AP-MS were overlapped. These two analyses further raise the concern for the quality of the current study and should be further explored.

6. Line 139-140, "As BioID can capture transient and proximal interactions, most experimental validation methods, such as coimmunoprecipitation, are not sufficiently sensitive to validate the results". This statement is not accurate and also leads to the major problem of this study – none of the novel interaction was properly validated by alternative approaches, such as coimmunoprecipitation for stable interaction and proximity interaction by immunofluorescence colocalization or reciprocal proximity labeling and western blot. The data presented in Fig. 4C-D presented nice functional exploration of the NFIA related interaction hub. However, the interaction should be firstly validated. The authors should provide multiple cases of validation especially for the extensively discussed prey-prey correlation analysis part.

Minor comments:

1. In general, the description for figure legends should be properly extended by including critical information. For example, it's not clear that the western blot in Fig. 4C is properly repeated.

2. The information in Fig. 1D and Fig. 2B is largely redundant and should be improved.

3. Fig. 2A is too busy to clearly present the overlap between AP-MS and BioID profiling of the same bait proteins. Better presentation of this critical information should be provided.

4. Line 111, "The BioID method has been suggested to be efficient for studying transient interactions, and this is supported by our results, strongly suggesting that BioID is the method-of-choice for studying TF". The authors should specify how their own results could support this statement.

5. There are some typos which need to be carefully corrected. For example, line 184.

6. In several figures, it's not clear whether BioID data, AP-MS data or combined interaction network are presented and discussed.

Reviewer #3:

Remarks to the Author:

In the paper by Göös et al. the authors examine the interactome of 110 selected transcription factors (TFs) when misexpressed in HEK 293 cells. Using two mass-spectrometry based methods to characterize stable (AP-MS) and more transient (BioID) protein-protein interactions the authors detect thousands of interactions, of which many have not yet been reported. The authors perform several clustering analyses and show that TFs within the detected interactomes often consist of TFs of the NFI family. To examine if NFI TFs may regulate the activity of the analyzed TFs the authors perform Luciferase-reporter assays. These experiments show that NFIA knock-down affects the activity of the examined TFs, regardless of their interaction with NFIA. Furthermore, using ChIP-seq the authors further show that the binding pattern of ectopically expressed SOX2 is significantly affected by NFIA knock-down. Although this study addresses an important question the ectopic context that the experiments are performed in questions their biological relevance.

Comments:

1)The experiments are performed in HEK293 cells. This cellular context is highly ectopic for some of the TFs. To what extent is the cellular context and growth conditions of the HEK293 cells affecting the composition of the reported interactomes? The authors must discuss the value of analyzing TF interactomes in an ectopic context?

2)The authors show that upon the removal of NFIA, the genome-wide binding pattern of SOX2 is substantially altered. These experiments raise a number of questions.

a)What antibodies were used to precipitate SOX2?

b)Is NFIA binding overlapping with that of SOX2?

c)How many times were these experiments repeated?

d)What TF binding motifs were enriched in the SOX2 bound regions (peaks)?

e)Is there a selective enrichment of NFI binding motifs in SOX2 bound regions sensitive to NFIA levels?

3)Using luciferase-reporter assays the authors examine how TF function is affected by the loss of NFIA. Are the regulatory DNA regions containing NFI binding motifs? If so, what is the distance between binding sites of the examined TFs? Most TFs appear to be affected by the loss of NFIA, even TFs that are not interacting with NFIA. The authors conclude that this is due to an indirect regulation of their activities. Why are not these important controls shown in the main figure? Why are only synthetic regulatory DNA-regions analyzed?

Reviewer #4:

Remarks to the Author:

The manuscript entitled "Human transcription factor protein interaction networks" describes an extensive characterization of PPI networks of 110 transcription factors using AP-MS and BioID. The paper thus provides an extensive list of several thousand potential interactions.

Two approaches are included. The first one is using the single-step Strep-tag affinity purification for AP-MS and the second one is using the same system but to purify proteins that were biotinylated by BirA* following addition of biotin to the media. It is mentioned that the expression of studied TF was adjusted to close to physiological level of the Tet inducible system, but no experimental evidence is presented in the paper. Considering the low expression level of TF in general, I doubt this was possible to achieve using the Flp-In Tet inducible system, which is notoriously difficult to titrate.

Regarding the large amount of data and the number of PPI identified, I find it relatively troubling that such a large proportion of interactions (over 75%) were not previously reported. Other large-scale studies with TF published have very little overlap with their own, as mentioned in the text. There are also several studies with transcriptions factors that have been performed using BioID that could have been used to compare with their own BioID, which has identified the most interactions. This is hand-waved on the fact that the difference is likely due to the transient nature of the TF interactions and/or the different tagging strategies used, but in the absence of validation, it is difficult to determine which of the studies (theirs or the others) represent a true catalogue of PPI for TFs.

Overall, the manuscript is interesting and includes a large amount of data. The following analysis performed underlines interesting observations, including the interactions with other transcription factors and the identification of biological complexes. However, the data presented in a large number of tables makes the data somewhat hard to digest.

Other comments:

The identified proteins were filtered using a SAINT score cut-off of 0.74, but no further discussion is provided on why this value was used. Considering the analysis presented the results from 2-4 repeats for each TF, I think it would be appropriate to provide a better rationale. Is this threshold actually stringent, and are they losing significant proteins that should be taken into account?

Some important information is missing for protein identification with Proteome Discoverer. An FDR for peptide was set to 0.05. What was the minimum number of peptides used in the study? Was that taken into accounts when identifying an interaction? For example, an identification in at least 2 of the repeats could be used to filter out potential contaminants.

The quality of the text is poor. There are several errors that goes beyond typos or grammatical errors. The text will have to be extensively edited for English.

REVIEWER COMMENTS

Reviewer #1 (Expertise: MS-proteomics for systems-level cell biology):

In their paper, Goos et al. describe their efforts to map protein interactions for 110 human transcription factors using both AP-MS and BioID. The result of this effort is a sizeable interaction network, most of which has not been previously described, and which provides numerous insights into the biology of individual transcription factors and transcription complex organization more generally. The authors select a couple of specific examples for targeted follow-up, looking for effects of NFIA silencing on activity of several transcription factors to which it binds. They also used ChIP-seq to show that NFIA silencing impacts SOX2 binding to chromatin. Finally, they mine their network for additional biological insights using various bioinformatic approaches, several of which they discuss in some detail. While I expect this work will eventually be of broad interest to the scientific community, I have some technical concerns and questions that need to be addressed before this paper will be ready for publication.

MAJOR COMMENTS

1. As the authors point out, there are ~1600 human transcription factors. Since just a fraction of these (110) were characterized in this study, how these were selected and how representative they are of the larger group will have important implications for how this network can be used. How did the authors decide which transcription factors to target in this study?

The selection process is now clarified in beginning of the Results-sections (Page 5, rows 90-100): “To systematically investigate the protein–protein interactions of human TFs, we selected a representative set of 109 TF genes from different TF families (Table S1A). Selection was based on the availability of full-length TF constructs. Selected TFs were analyzed in two biological replicates and, as the correlation between the technical and biological replicates were excellent (Figure S1A), either in one or two technical replicates. TFs are often classified according to their DNA-binding domains (DBDs), and the DBD distribution of studied TFs compared to all human TFs is shown in Figure 1A. The majority of the studied TFs had C2H2 zing finger (ZF) or homeodomain DBDs, which are the most common DBDs among the human TFs⁴.

The selected TFs were subjected to two independent mass spectrometry-based interactome analysis methods (Figure 1B).”

2. Close examination of Table S1B-C raises some questions about how the data were filtered to identify significant interactions. First, in both BioID and AP-MS datasets, the target baits are themselves reported as interacting proteins. For example, at the top of Table S1C we find the protein BRAC listed in the “Prey” column for the BRAC pulldown. It’s also listed as a prey when it is the target of BioID analysis as well (Table S1B). This is true for most or all baits in both AP-MS and BioID datasets. That these bait proteins are detected – and strongly enriched – is to be expected, since they are being specifically targeted for purification. However, labeling these proteins as “preys” in their own IP’s and including these observations in final interaction lists and total interaction counts for the dataset as the authors have done is misleading. People accessing these data in the future may erroneously take these entries to imply homo-dimerization, especially once these data are incorporated into databases like BioGRID and IntACT. These entries should be

removed from the supplementary tables and the total interaction counts throughout the text adjusted accordingly.

We thank the reviewer pointing out this important point. This is ongoing debate in interactomics, but bait-bait interactions are commonly reported – as analysing the same organism the lack of these is not possible to prove. This feature is visible in all of the data repositories as well (there are over 8000 “self-interactions” in BioGrid, over 2500 in IntAct, and nearly 1000 in PINA2).

However, as suggested by the reviewer all bait-bait interactions are now removed, and figures and tables corrected accordingly.

3. A second concern emerging from examination of Table S1B is that at least one obvious contaminant protein seems to be detected repeatedly as an interacting partner. Bovine serum albumin (BSA) from cell culture media is identified as an interacting partner via BioID for 11 different transcription factors, representing 10% of the dataset: ELK4, FOS, FOXQ1, GATA3, GLI2, HME1, LHX6, MYC, MYOD1, SMAD5, and STAT3. These are only detected in the BioID dataset. I didn't find BSA listed at all in the AP-MS interaction list. This raises a few points. First, the authors should filter out these interactions since they're clearly artifacts. More generally, if BSA passed as an interacting protein in 10% of their BioID experiments, this raises questions about whether their BioID platform is particularly susceptible to interference from background proteins and to what extent other contaminating proteins remain in their filtered interaction lists. Are the authors sure they've sufficiently dealt with these artifacts?

Bovine albumin is now removed from the list. In addition, we used control purification data from CRAPome-database (that includes contaminant proteins with abundancies and frequencies from 716 experiments) to further filter possible contaminants from the high-confidence interactor list.

In BioID data, only the proteins with lower frequency than 50 % (358/716) in CRAPome were allowed. Also preys with CRAPome frequency 25-50% (179-358/716) and with higher CRAPome average spectral count compared to our average spectral count were removed. This led to exclusion of 520 interactions from BioID data (Table S1B).

From AP-MS data, in addition to Saint cut-off, preys that were present with higher frequency than 50% in Crapome (358/716) were required to have three-time higher average spectral count than average spectral count in CRAPome. This led to exclusion of 640 interactions from AP-MS data (Table S1C).

As the BioID and AP-MS methods differ from each other in interaction stability, the exactly same filtering methods cannot not be utilized as it leads to not stringent enough filtering of BioID data. However, we are confident that now with the strict filtering used, the resulting interactions are actually very high-confidence interactions. Tables, figures and text are now corrected accordingly.

4. Starting on line 244, the authors highlight a cluster of actin/myosin-related prey proteins in the BioID dataset that cluster together and link to several transcription factors. The authors interpret these interactions as evidence of a link between these transcription factors (especially FOS and STAT1) and actin/myosin signaling in the nucleus. Perhaps this is the case - there is evidence that some of these proteins (ACTB, MYO1C, etc) do reside in the

nucleus where they perform functions related to chromatin remodeling and transcription regulation. However, a skeptic might wonder whether most of these actin/myosin-related proteins are instead artifacts for several reasons. First, though a few of these proteins are known to reside partially in the nucleus and participate in transcription-related processes, many are not. (Examples include MYO1B, MYO1D, MYL6B, MYH9, MYH10, MYO5A, etc. according to Uniprot and Human Protein Atlas as listed in Table S2 of this paper.) Second, actin, myosin, and related proteins are themselves abundant and are often detected as background in these kinds of experiments. Third, there may in fact be elevated background in the BioID experiments targeting baits associated with Cluster 2. As I mentioned above, the contaminant BSA is present at elevated levels and flagged as an interacting partner in BioID experiments associated with 11 baits, reflecting anomalous background. Nine of these baits are linked to this actin/myosin cluster (Cluster 2 - see Table S7). Fourth, relatively few of the preys in this cluster appear to be nuclear, while several (PGRC1, TIM50, RAB8A, ATPA, ATPB, AT1A1, ADT2, ENPL, CALX, etc) localize to membranes, mitochondria, ER, and other alternate compartments and would seem to be contaminants (Table S7). Finally, while the BioID data for FOS and STAT1 reveal many proximal actin/myosin proteins, none of these interactions appear in the AP-MS data. While this could mean that these interactions are too transient or weak to survive AP-MS purification, it could also mean that these are artifacts specific to these BioID samples, much like BSA and the membrane/mitochondrial/ER preys detected alongside them. If the authors want to highlight this cluster, they will need to make a stronger case that these observations represent biology and should provide additional independent experimental confirmation to show that these links between FOS/STAT1 and actin/myosin proteins are real. Alternatively, they could cut this example and focus instead on other clusters where the data are more convincing.

We thank the reviewer for the critical aspects of the actin and myosin related cluster. We agree that these actin/myosin proteins are difficult to study as they are frequently observed as a high abundant contaminant. However, transcription factors FOS and STAT1 do seem to have a very specific actin- and myosin- linked interactions. They for example uniquely interacted with ACTB and ACTBL2 within studied 109 TFs. However, due to the reviewer's listed reasons and the scope of this article, we have as suggested decided to leave this cluster out as a highlight.

5. I also note that the authors report that the mitochondrial carboxylase ACACB interacts with about 22 different transcription factors. This is also likely an artifact, as this protein is endogenously biotinylated according to Uniprot and Lee et al. (2008) *Proteins* 72:613-24. I should also point out that all 22 of these IP's with excessive ACACB levels are associated with Cluster 2, seemingly linking that cluster to another experimental artifact. Endogenous biotinylation may also explain the presence of some other likely artifacts reported as interactions in the BioID data, including mitochondrial carboxylases MCCA, MCCB, PCCA, and PCCB that are covalently modified with biotin (e.g. Samavarchi-Tehrani et al. (2020) *Mol Cell Proteom* 19:757-773).

We agree with the reviewer on this. It is often a difficult task to remove all probable contaminants using a systematic filtering strategy, and ACACB is seen as an interactor in many articles. However, using now the approach with more stringent filtering parameters (including the usage of CRAPome) (described in detail earlier (Q3)), complete removal of ACACB and in most cases of all the mitochondrial carboxylases was achieved.

6. In various places in the text the authors contrast BioID and AP-MS, saying for example that BioID "... was used to detect also transient and proximal interactions of the TFs." (Line 96). And "The BioID method has been suggested to be efficient for studying transient interactions..." (Line 111). These statements are true, though the authors should point out explicitly that as long as they're close together, a prey protein doesn't actually need to physically associate with a given bait to be labeled via BioID. And given the confined nature of chromatin, these TF's might be unusually prone to labeling other proteins that are in close proximity but don't actually interact.

We agree with the reviewer on this. It is often impossible to derive from BioID data alone if the interaction is stable, transient, direct or proximal. To clarify this, we have now added (lines 103-106) "Activation of BirA allows it to biotinylate proteins within close proximity (10 nm) of the studied TFs, including transient interactions. However, no physical contact is needed for biotinylation, and because of the confined nature of chromatin, proteins other than interacting proteins might also be labeled in low amounts. "

MINOR COMMENTS

1. I'm curious about the authors' observation that relatively few of the interactions they saw had been previously reported (Figure 1C). Is this fraction higher for really well characterized transcription factors (e.g. P53, MYC, etc) versus other less thoroughly characterized transcription factors?

Yes, well characterized TFs had more previously reported PPIs. It is now clarified in text (lines 165-168) "The PPIs of several well-studied TFs, such as SOX2, MYC, TYY1, PAX6, HNF4a and GATA2, overlapped with more than 45 known interactions in the databases, whereas the PPIs of other less studied TFs, such as ZIC3, ELK4, ESR1, IRF3 and IRF9, did not overlap with any known PPIs from the databases or studies (Table S1B)."

2. Figure 1D: It looks like about 20 baits are missing from this plot. Why aren't all 110 shown? If the authors can't fit them all in the plot, then they should at least indicate in the figure legend that only a subset of baits are shown. Also, there seems to be a typo – one bait is simply labeled as "T".

Thank you for noticing that graphical issue. Scaling of the figure had resulted that some of the baits were not properly visible. Now all the 109 baits are visible in the figure. T-box transcription factor T "T" is also now for clarity changed to BRAC, which is the alternative name.

3. Line 140: I'm not sure it's fair to say co-IP isn't sensitive enough. The issue is that AP-MS and BioID are measuring two phenomena that are related but distinct. One measures physical binding while the other measures proximity.

We thank the reviewer for this comment, and being the catalyst for us performing this experiment in large scale. We now agree with the reviewer that this actually is feasible, as we obtained high-quality (>90%, 65/70 of the tested interaction pairs) validation data for NFI family of transcription factors and their interactors using co-IP approach. As it is possible that the approach does not work as well with all transcription factors, we have toned down the sentence to state "As BioID can capture transient and proximal interactions, most

experimental validation methods, such as coimmunoprecipitation, might not be sensitive enough for validation of the results” (rows 147-148).

4. Figure 2: I’m having trouble distinguishing the blue and black edges. I’d change the color scheme for improved contrast. The same is true for Figure 4A.

We thank the reviewer pointing this out. We have now changed the black edges to green to improve the contrast.

5. Figure 4C. It seems that HME1 is an alternative name for EN1 (both labels are given in Figure 2). Why is there a row labeled “HME1” in the PAX6 plot? This looks like a mislabeling. Also, the authors should use consistent terminology in the panel labeled EN1 (Lower lefthand corner of Figure 4C) – right now the same protein is labeled EN1 at the top of the plot and HME1 below the bar graph.

We thank the reviewer for noticing this. We have now changed EN1 to HME1 everywhere and mislabeling of PAX6 plot is corrected.

6. In the Methods section titled “Protein identification” the authors write “FASTA library was complemented with BSA, tag sequences, trypsin, biotin, and GFP.” Since biotin is a small molecule, perhaps the authors mean they included streptavidin (in place of strep-tactin, whose sequence is presumably proprietary)?

Biotin is now removed from the text. Thank you for pointing this mistake/typo.

7. In the “Protein identification” section the authors also write that “oxidation ... of methionine or N-terminus were used as dynamic modification”. Oxidation of the N-terminus doesn’t make chemical sense in this context. Do the authors mean acetylation of the N-terminus? Also, what was the product ion mass error?

N-terminus was added into sentence accidentally and is now removed from the text. We have also added details about product mass error: “The precursor mass tolerance was set to 15 ppm, and the fragment mass tolerance was set to 0.05 Da” (lines 530-31)

8. Instrumentation Methods (lines 522 – 533): What dynamic exclusion settings were used on the instrument?

Settings are now added to text: “Finally, dynamic exclusion was enabled and settings were set as follows: repeat count: 1, repeat duration: 30s, exclusion list size: 500, exclusion duration 30s, exclusion mass width relative to 5 ppm.

9. In line 544 the authors write that “All the TFs were analyzed in two or four replicates”. What determined whether each TF was analyzed two versus four times? Were these biological or technical replicates? Were replicates performed separately or side-by-side?

In the beginning of the project, two technical replicates were produced from each biological replicate. However, we soon detected, that variability between the technical replicates was very low, and therefore chose to only run one technical replicate out of each biological replicate. We’ve now described this via a correlation plot Figure S1A: technical replicates

had a correlation value of 0.974 and biological replicates 0.970, suggesting insights gained from technical replicates were negligible.

This is now corrected and reads in text (lines 92-94) “Selected TFs were analyzed in two biological replicates and, as the correlation between the technical and biological replicates were excellent (Figure S1A), either in one or two technical replicates.”

10. Line 542: the authors say they filtered peptides to an FDR <5%. What was the resulting protein-level FDR? And what additional steps (if any) were taken to control the protein-level FDR?

We did not calculate the global protein-level FDR for the unfiltered interactome data. However, on the SAINT filtering average spectral count cut-off was set to ≥ 2 (detected in $\geq 3/4$ of the samples). This results to high-confidence interacting proteins to be assigned with the very stringent two-peptide rule.

11. Line 546. The authors write “...we used 44 and 75 similarly tagged and analysed GFP control runs for the BioID analysis and AP-MS analysis, respectively. We also included GFP’s with a nuclear localization signal...” Are the authors saying that those 44 and 75 GFP control runs in BioID and AP-MS were performed using GFP with a nuclear localization signal? Or were the analyses of GFP’s with a nuclear localization signal separate from those listed in the preceding sentence?

This is now clarified better in the text (lines 569-574) “To efficiently filter the real interactions, we used 44 and 75 similarly tagged and analyzed GFP control runs for the BioID analysis and AP-MS analysis, respectively. From these, 16 and 18, respectively, had a nuclear localization signal (NLS) to efficiently filter out nonspecific nuclear interactions. All GFP controls were used as negative controls in SAINT analysis, where the large nuclear dataset further facilitated the frequency-based deletion of contaminating proteins.”

Reviewer #2 (Expertise: MS-based proteomics):

The manuscript by Göös et al. with title of “Human transcription factor protein interaction networks” presented a comprehensive interactome profiling of 110 human transcription factors using both AP-MS and BioID approaches. As the authors well summarized that transcription factors represent 6-9% of human proteome and play crucial roles in gene transcription related mechanisms. A systematic profiling of such type of protein machinery is desirable and should provide a valuable resource to related biological fields. The combination of traditional AP-MS analysis in cell lysate and proximity labeling-proteomic analysis in living cells should provide complimentary information for better viewing their molecular functionality. However, current manuscript didn't present sufficient information to support the above goals and therefore needs to be significantly improved before further consideration.

Major comments:

1. Interactome profiling of transcription factors has been a popular topic in the past decade especially by using standard AP-MS approach. However, the authors failed to systematically review and summary these early efforts and clearly present the novelty of their own study. For example, yeast chromatin-associated interactome by Figeys and co-workers in 2010 (Mol. Syst. Biol., 6, 448) and large-scale endogenous IP-MS analysis of transcriptional proteins by Qin and co-workers in 2011 (Cell, 145, 787). In addition, continuous global-scale AP-MS analysis of more than 10,000 human proteins by the BioPlex project leading by Gygi and co-workers should also cover a significant fraction of transcription factors (BioRxiv, <https://doi.org/10.1101/2020.01.19.905109>). Except for expanding early efforts for profiling transcription factor interactome, the authors should deliver more clear biological rationale for such type of study and major biological conclusion.

We thank the reviewer for suggesting these large studies to be compared to our interactions. The overlap between the interactions identified in these studies and our study is now presented in Table S1. We additionally now analyzed the overlap between the three transcription factor interactome papers, BioPlex (which also included many TFs), and our dataset (Figure S1B). The overlap between our AP-MS data and Li et al (PMID25609649), Qin et al (PMID21620140), and Lambert et al (PMID21179020) were 2.34, 0.96, 1.43, respectively. The overlap between the aforementioned three TF interactomes ranged from 0 to 1.92 %. For our BioID interactomes data, the overlap ranged between 0.25 % and 1.56 %. For these pairwise comparisons, we only used baits included in both interactomes. With BioPlex, the overlap was 2.92 % for our AP-MS data, 0.81 % for our BioID data, and between 0.05 and 4.17 % for the three previously published datasets. Additionally, we have added discussion of this to text (lines 168-173).

2. Along this line, the authors didn't explain why the 110 transcription factors were selected for this specific study. Are there any previous AP-MS studies which have studies some of them? The authors should perform a related bioinformatic analysis and provide extended discussion about the rationale for studying this limited fraction of transcription factors.

The selection process is now clarified in beginning of the Results-sections (Page 5, rows 90-100): “To systematically investigate the protein–protein interactions of human TFs, we selected a representative set of 109 TF genes from different TF families (Table S1A). Selection was based on the availability of full-length TF constructs. Selected TFs were

analyzed in two biological replicates and, as the correlation between the technical and biological replicates were excellent (Figure S1A), either in one or two technical replicates. TFs are often classified according to their DNA-binding domains (DBDs), and the DBD distribution of studied TFs compared to all human TFs is shown in Figure 1A. The majority of the studied TFs had C2H2 zinc finger (ZF) or homeodomain DBDs, which are the most common DBDs among the human TFs⁴.

The selected TFs were subjected to two independent mass spectrometry-based interactome analysis methods (Figure 1B).”

3. As a valuable resource, the data quality is critically important especially for BioID type of data which represent the proximal proteome and tends to generate significant amount of nonspecific labeling.

The authors only presented very general overview of the design of the study. Current data presentation failed to convince the audience. The authors have to significantly expand their data analysis dedicated for data quality part. Specifically, for example, the authors mentioned two or four replicates were performed. Why it is not consistent for such a critical experimental design? How the data reproducibility was evaluated and controlled? The authors used 44 and 75 GFP control runs for the BioID analysis and AP-MS analysis, respectively. How did these important control data use for filtering out nonspecific adsorption and labeling? For all the figures in this study, the authors only presented the interaction network data. It’s hard to conclude the reliability of the highlighted interaction hubs – associated peptide counts etc. should be properly presented.

We apologize for not obviously providing enough information about study design, controls and data filtering. Partly this was due to the strict space limits.

In the beginning of the project, two technical replicates were produced from each biological replicate. However, we soon detected, that variability between the technical replicates was very low, and therefore chose to only run one technical replicate out of each biological replicate. We’ve now described this via a correlation plot Figure S1A: technical replicates had a correlation value of 0.974 and biological replicates 0.970, suggesting insights gained from technical replicates were negligible.

This is now corrected and reads in text (lines 92-94) “Selected TFs were analyzed in two biological replicates and, as the correlation between the technical and biological replicates were excellent (Figure S1A), either in one or two technical replicates.”

Regarding the GFP controls. this is now clarified better in the text (lines 569-574) “To efficiently filter the real interactions, we used 44 and 75 similarly tagged and analyzed GFP control runs for the BioID analysis and AP-MS analysis, respectively. From these, 16 and 18, respectively, had a nuclear localization signal (NLS) to efficiently filter out nonspecific nuclear interactions. All GFP controls were used as negative controls in SAINT analysis, where the large nuclear dataset further facilitated the frequency-based deletion of contaminating proteins.”

To enhance the filtering even further, we now used additional control purification data from CRAPome-database (that includes contaminant proteins with abundancies and frequencies from 716 experiments) to further filter possible contaminants from the high-confidence interactor list.

In BioID data, only the proteins with lower frequency than 50 % (358/716) in CRAPome were allowed. Also preys with CRAPome frequency 25-50% (179-358/716) and with higher CRAPome average spectral count compared to our average spectral count were removed. This led to exclusion of 520 interactions from BioID data (Table S1B).

From AP-MS data, in addition to SAINT cut-off, preys that were present with higher frequency than 50% in Crapome (358/716) were required to have three-time higher average spectral count than average spectral count in CRAPome. This led to exclusion of 640 interactions from AP-MS data (Table S1C).

As the BioID and AP-MS methods differ from each other in interaction stability, the exactly same filtering methods cannot not be utilized as it leads to not stringent enough filtering of BioID data. However, we are confident that now with the strict filtering used, the resulting interactions are actually very high-confidence interactions. Tables, figures and text are now corrected accordingly.

We thank the reviewer for suggesting adding the peptide counts and other details also in figures. We tried this, but for our and colleagues' opinion this resulted in extremely busy and difficult figures, and we therefore decided to leave these details to table format (Table S1)

4. In Fig. 1C, the authors highlighted much higher number of identified PPI by the BioID approach. As stated before and the cited papers by the authors, it's not proper to define the identified proteins as PPI, but only proximal proteins. Along this line, the authors should extend their discussion about how the nonspecific labeling was properly filtered? Any more accurate quantification was used, which has becoming the standard approach in the proximity proteomics field.

We, and many leading laboratories performing BioID analyses use SAINT (PMID: 24513533) for statistically filtering the high-confident interactions. This approach has proven to be suitable for even with large-scale data with variable bait localizations (PMID: 34079125). The SAINT filtering uses spectral counting. Additionally, our control set used in SAINT is larger than mostly used by others and includes tagged GFP controls. The set includes 28 (BioID) and 57 (AP-MS) normal GFP analyses (mainly cytoplasmic localization) and 16 (BioID) and 18 (AP-MS) nuclear localized NLS-GFP controls. This and the use of the new CRAPome (citation) database with 716 controls allowed us now to perform extremely stringent data filtering. Of note is that using the BioID tag allows much better control of the biotinylation than the newer much more efficient version of the BirA or APEX approaches. With these approaches the data shows much higher background and e.g. batch to batch variation, making the data extremely challenging to filter.

In respect to MS1 quantification, we have not performed this as this approach is not compatible with the SAINT filtering approach. Naturally these values would be possible to obtain from our data if needed, however, we think this would not add much to this article. If there would be a clear idea / analysis type which would utilize this data we would be glad to include this.

5. Line 104, the authors highlighted >75% TF interactome were previously unreported. The authors should specify the resource for such a comparison. In the section of "Comparison to other studies", the authors well compared with Li et al.'s work for studying 59 TF interactome. However, only 6% of PPIs identified by BioID and 26% of PPIs by AP-MS

were overlapped. These two analyses further raise the concern for the quality of the current study and should be further explored.

We have now improved our comparison to several other studies. Apart from Li et al. study, the identified PPIs are now overlapped with public interaction databases such as PINA2, STRING, IntAct and Biogrid and several medium-to-large –scale interactome studies such as Lambert et al., Malovannaya et al., and Huttlin et al. (Tables S1B-C).

We additionally now analyzed the overlap between the three transcription factor interactome papers, BioPlex (which also included many TFs), and our dataset (Figure S1B). The overlap between our AP-MS data and Li et al (PMID25609649), Qin et al (PMID21620140), and Lambert et al (PMID21179020) were 2.34, 0.96, 1.43, respectively. The overlap between the aforementioned three TF interactomes ranged from 0 to 1.92 %. For our BioID interactomes data, the overlap ranged between 0.25 % and 1.56 %. For these pairwise comparisons, we only used baits included in both interactomes. With BioPlex, the overlap was 2.92 % for our AP-MS data, 0.81 % for our BioID data, and between 0.05 and 4.17 % for the three previously published datasets. Additionally, we have added discussion of this to text (lines 168-173).

6. Line 139-140, “As BioID can capture transient and proximal interactions, most experimental validation methods, such as coimmunoprecipitation, are not sufficiently sensitive to validate the results”. This statement is not accurate and also leads to the major problem of this study – none of the novel interaction was properly validated by alternative approaches, such as coimmunoprecipitation for stable interaction and proximity interaction by immunofluorescence colocalization or reciprocal proximity labeling and western blot. The data presented in Fig. 4C-D presented nice functional exploration of the NFIA related interaction hub. However, the interaction should be firstly validated. The authors should provide multiple cases of validation especially for the extensively discussed prey-prey correlation analysis part.

We now agree with the reviewer on this. And we thank the reviewer for suggesting validation on the “nice functional” NFIs interactions hubs.

We now obtained high-quality validation data for NFI family of transcription factors and their interactors. We have now added extensive co-IP validation for the NFI family members as suggested by the reviewers. This validation resulted of a large portion (>90%) of the MS detected interaction pairs (65/70) and in total 99/111 of the tested interaction prey pairs to be validated (this is added in results (lines 221-235), Supplementary Figure S3, and materials & methods), significantly strengthening the validity and biological relevance of our identified interactions. We also have accordingly adjusted our initial claims that the validation of the identified interactions might be difficult with co-IP due to possibly transient nature of the interactions. However, as it is possible that the approach does not work as well with all transcription factors, we have toned down the sentence and now state (lines 147-148)“As BioID can capture transient and proximal interactions, most experimental validation methods, such as coimmunoprecipitation, might not be sensitive enough for validation of the results”

Minor comments:

1. In general, the description for figure legends should be properly extended by including

critical information. For example, it's not clear that the western blot in Fig. 4C is properly repeated.

We have now included more description to the figure legends. All experiments have been performed at least with two replicate, and for qualitative data a representative experiment is shown. Uncropped blot of the representative experiments of the WB (Fig. 4C) is added to the source data table.

2. The information in Fig. 1D and Fig. 2B is largely redundant and should be improved.

Figure 1F and Figure 2B to our opinion describe somewhat different things. Figure 2B tries to transmit the similarities within the TF families, whereas the figure 1F is showing individual TFs interactions and act also as initial glimpse to the data together.

3. Fig. 2A is too busy to clearly present the overlap between AP-MS and BioID profiling of the same bait proteins. Better presentation of this critical information should be provided.

We agree that figure Figure 2A was overly busy to see the overlap between AP-MS and BioID. We have now modified Figure 2A and added new figure 1C, where the overlap is more clearly shown.

4. Line 111, "The BioID method has been suggested to be efficient for studying transient interactions, and this is supported by our results, strongly suggesting that BioID is the method-of-choice for studying TF". The authors should specify how their own results could support this statement.

We believe that the majority of the results actually support this statement. The BioID results to large number of biologically relevant interactions. If we would need to pick one example, this is most strongly supported by the Gene Ontology analysis of the interactors detected by the BioID in Figure 3A. Additionally the prey-prey analysis identified several biologically relevant clusters (Figure 5).

5. There are some typos which need to be carefully corrected. For example, line 184.

We apologize for the typos. We have now used a Nature publishing group's professional scientific language (English) editing service to improve grammar and language. We believe this has significantly improved the quality of the text.

6. In several figures, it's not clear whether BioID data, AP-MS data or combined interaction network are presented and discussed.

We apologize this unclarity and have now added more clearly indication in figure legends of which data is used for which figure.

Reviewer #3 (Expertise: Sox transcription factors, ChipSeq):

In the paper by Göös et al. the authors examine the interactome of 110 selected transcription factors (TFs) when misexpressed in HEK 293 cells. Using two mass-spectrometry based methods to characterize stable (AP-MS) and more transient (BioID) protein-protein interactions the authors detect thousands of interactions, of which many have not yet been reported. The authors perform several clustering analyses and show that TFs within the detected interactomes often consist of TFs of the NFI family. To examine if NFI TFs may regulate the activity of the analyzed TFs the authors perform Luciferase-reporter assays. These experiments show that NFIA knock-down affects the activity of the examined TFs, regardless of their interaction with NFIA. Furthermore, using ChIP-seq the authors further show that the binding pattern of ectopically expressed SOX2 is significantly affected by NFIA knock-down. Although this study addresses an important question the ectopic context that the experiments are performed in questions their biological relevance.

Comments:

1)The experiments are performed in HEK293 cells. This cellular context is highly ectopic for some of the TFs. To what extent is the cellular context and growth conditions of the HEK293 cells affecting the composition of the reported interactomes? The authors must discuss the value of analyzing TF interactomes in an ectopic context?

We thank the reviewer for this comment and agreed that we should discuss the value of analyzing the TF interactomes in HEK 293 cells. We have added the following sentences in the discussion part (lines 477-483): “From the established and used cell lines, HEK293 express a wide range of different transcript, totaling ~ 13k genes. In interaction proteomics HEK293 cells have been widely used as a “gold standard” for studying protein-protein interactions (PPIs) (Taipale et al., 2015 (PMID: 25036637); Huttlin et al., 2015 (PMID: 26186194); Huttlin et al., 2021 (PMID: 33961781)). The proteome functional classification of HEK293 mimicks UniProt and most closely resembled the distribution observed for preys (Huttlin et al., 2015 (PMID: 26186194)). Previous study has shown that even with the lentiviral infection of 293 cells, overexpression has little effect on identification of true interacting partners after statistical filtering (Liu et al., 2020 (PMID: 32778839)).”

2)The authors show that upon the removal of NFIA, the genome-wide binding pattern of SOX2 is substantially altered. These experiments raise a number of questions.

a)What antibodies were used to precipitate SOX2?

The antibody we used to precipitate SOX2 is against HA tag on SOX2 (Abcam, ab18181, Lot No: GR3288712).

b)Is NFIA binding overlapping with that of SOX2?

Yes, we have performed two independent experiments and found NFIA ChIP-Seq has 129 peaks (experiment 1), and 195 peaks (experiment 2), respectively, overlapping with SOX2. The results shown as below:

c) How many times were these experiments repeated?

We have performed two independent experiments and obtained comparable results, as shown in the following figures:

Experiment 1

Experiments 1 and 2: Heatmap representation of SOX2 binding intensity based on ChIP-seq signals in 293T cells while treated with Control siRNA and the siRNA against NFIA, respectively.

d)What TF binding motifs were enriched in the SOX2 bound regions (peaks)?

TF binding motifs of the SOX family members, including SOX3, SOX2, SOX21 and so on indicated most enriched in the SOX2 ChIP-seq peaks. The top enriched TF binding motifs were shown as below:

Experiment 1, SOX2 ChIP-Seq motif

Rank	Motif	Name	P-value	log P-pvalue	q-value (Benjamini)
1		Sox3(HMG)/NPC-Sox3-ChIP-Seq(GSE33059)/Homer	1e-325	-7.485e+02	0.0000
2		Sox2(HMG)/mES-Sox2-ChIP-Seq(GSE11431)/Homer	1e-277	-6.382e+02	0.0000
3		Sox21(HMG)/ESC-SOX21-ChIP-Seq(GSE110505)/Homer	1e-272	-6.281e+02	0.0000
4		Sox10(HMG)/SciaticNerve-Sox3-ChIP-Seq(GSE35132)/Homer	1e-258	-5.946e+02	0.0000
5		Sox15(HMG)/CPA-Sox15-ChIP-Seq(GSE62909)/Homer	1e-256	-5.905e+02	0.0000
6		Sox6(HMG)/Myotubes-Sox6-ChIP-Seq(GSE32627)/Homer	1e-224	-5.172e+02	0.0000
7		Sox17(HMG)/Endoderm-Sox17-ChIP-Seq(GSE61475)/Homer	1e-209	-4.830e+02	0.0000
8		Sox4(HMG)/proB-Sox4-ChIP-Seq(GSE50066)/Homer	1e-175	-4.040e+02	0.0000
9		CDX4(Homeobox)/ZebrafishEmbryos-Cdx4.Myc-ChIP-Seq(GSE48254)/Homer	1e-157	-3.635e+02	0.0000
10		HOXB13(Homeobox)/ProstateTumor-HOXB13-ChIP-Seq(GSE56288)/Homer	1e-152	-3.508e+02	0.0000
11		Sox7(HMG)/ESC-Sox7-ChIP-Seq(GSE133899)/Homer	1e-140	-3.232e+02	0.0000
12		Hoxa11(Homeobox)/ChickenMSG-Hoxa11.Flag-ChIP-Seq(GSE86088)/Homer	1e-104	-2.399e+02	0.0000
13		Sox9(HMG)/Limb-SOX9-ChIP-Seq(GSE73225)/Homer	1e-93	-2.148e+02	0.0000
14		Cdx2(Homeobox)/mES-Cdx2-ChIP-Seq(GSE14586)/Homer	1e-91	-2.117e+02	0.0000
15		Hoxd11(Homeobox)/ChickenMSG-Hoxd11.Flag-ChIP-Seq(GSE86088)/Homer	1e-91	-2.098e+02	0.0000

Experiment 2, SOX2 ChIP-Seq motif

Rank	Motif	Name	P-value	log P-value	q-value (Benjamini)
1		Sox3(HMG)/NPC-Sox3-ChIP-Seq(GSE33059)/Homer	1e-897	-2.067e+03	0.0000
2		Sox21(HMG)/ESC-SOX21-ChIP-Seq(GSE110505)/Homer	1e-837	-1.927e+03	0.0000
3		Sox2(HMG)/mES-Sox2-ChIP-Seq(GSE11431)/Homer	1e-771	-1.777e+03	0.0000
4		Sox10(HMG)/SciaticNerve-Sox3-ChIP-Seq(GSE35132)/Homer	1e-763	-1.759e+03	0.0000
5		Sox15(HMG)/CPA-Sox15-ChIP-Seq(GSE62909)/Homer	1e-700	-1.612e+03	0.0000
6		Sox6(HMG)/Myotubes-Sox6-ChIP-Seq(GSE32627)/Homer	1e-657	-1.514e+03	0.0000
7		Sox17(HMG)/Endoderm-Sox17-ChIP-Seq(GSE61475)/Homer	1e-611	-1.409e+03	0.0000
8		Sox4(HMG)/proB-Sox4-ChIP-Seq(GSE50066)/Homer	1e-484	-1.116e+03	0.0000
9		Fosl2(bZIP)/3T3L1-Fosl2-ChIP-Seq(GSE56872)/Homer	1e-377	-8.684e+02	0.0000
10		Jun-AP1(bZIP)/K562-cJun-ChIP-Seq(GSE31477)/Homer	1e-374	-8.618e+02	0.0000
11		Fra2(bZIP)/Striatum-Fra2-ChIP-Seq(GSE43429)/Homer	1e-373	-8.605e+02	0.0000
12		Fos(bZIP)/TSC-Fos-ChIP-Seq(GSE110950)/Homer	1e-373	-8.590e+02	0.0000
13		Fra1(bZIP)/BT549-Fra1-ChIP-Seq(GSE46166)/Homer	1e-366	-8.431e+02	0.0000
14		Sox7(HMG)/ESC-Sox7-ChIP-Seq(GSE133899)/Homer	1e-353	-8.139e+02	0.0000
15		JunB(bZIP)/DendriticCells-Junb-ChIP-Seq(GSE36099)/Homer	1e-347	-8.009e+02	0.0000

e)Is there a selective enrichment of NFI binding motifs in SOX2 bound regions sensitive to NFIA levels?

Overall, the answer is Yes. In the experiment 1, upon the NFIA knockdown, among the lost (6921) and common (362) SOX2 binding sites, we observed significant enrichment of NFI (NF1) binding motif. But we didn't observe NFI binding motif enriched among the gained (1341) peaks. The results of this analysis are shown as below:

The HOMER motif results of the lost 6921 SOX2 binding sites upon NFIA knockdown:

Rank	Motif	Name	P-value
1		Sox3(HMG)/NPC-Sox3-ChIP-Seq(GSE33059)/Homer	1e-269
2		Sox2(HMG)/mES-Sox2-ChIP-Seq(GSE11431)/Homer	1e-238
3		Sox21(HMG)/ESC-SOX21-ChIP-Seq(GSE110505)/Homer	1e-226
4		Sox15(HMG)/CPA-Sox15-ChIP-Seq(GSE62909)/Homer	1e-209
5		Sox10(HMG)/SciaticNerve-Sox3-ChIP-Seq(GSE35132)/Homer	1e-207
6		Sox6(HMG)/Myotubes-Sox6-ChIP-Seq(GSE32627)/Homer	1e-186
7		Sox17(HMG)/Endoderm-Sox17-ChIP-Seq(GSE61475)/Homer	1e-172
8		Sox4(HMG)/proB-Sox4-ChIP-Seq(GSE50066)/Homer	1e-147
9		HOXB13(Homeobox)/ProstateTumor-HOXB13-ChIP-Seq(GSE56288)/Homer	1e-138
10		CDX4(Homeobox)/ZebrafishEmbryos-Cdx4.Myc-ChIP-Seq(GSE48254)/Homer	1e-138
11		Sox7(HMG)/ESC-Sox7-ChIP-Seq(GSE133899)/Homer	1e-112
12		Hoxa11(Homeobox)/ChickenMSG-Hoxa11.Flag-ChIP-Seq(GSE86088)/Homer	1e-92
13		Sox9(HMG)/Limb-SOX9-ChIP-Seq(GSE73225)/Homer	1e-85
14		Cdx2(Homeobox)/mES-Cdx2-ChIP-Seq(GSE14586)/Homer	1e-85
15		Hoxd11(Homeobox)/ChickenMSG-Hoxd11.Flag-ChIP-Seq(GSE86088)/Homer	1e-77
16		Hoxd13(Homeobox)/ChickenMSG-Hoxd13.Flag-ChIP-Seq(GSE86088)/Homer	1e-74
17		Hoxd10(Homeobox)/ChickenMSG-Hoxd10.Flag-ChIP-Seq(GSE86088)/Homer	1e-73
18		Hoxa13(Homeobox)/ChickenMSG-Hoxa13.Flag-ChIP-Seq(GSE86088)/Homer	1e-71
19		Unknown(Homeobox)/Limb-p300-ChIP-Seq/Homer	1e-69
20		NF1(CTF)/LNCAP-NF1-ChIP-Seq(Unpublished)/Homer	1e-50
21		Zic3(Zf)/mES-Zic3-ChIP-Seq(GSE37889)/Homer	1e-49

The HOMER motif results of the 362 common binding sites:

Rank	Motif	Name	P-value
1		Sox3(HMG)/NPC-Sox3-ChIP-Seq(GSE33059)/Homer	1e-77
2		Sox21(HMG)/ESC-SOX21-ChIP-Seq(GSE110505)/Homer	1e-69
3		Sox10(HMG)/SciaticNerve-Sox3-ChIP-Seq(GSE35132)/Homer	1e-65
4		Sox2(HMG)/mES-Sox2-ChIP-Seq(GSE11431)/Homer	1e-58
5		Sox6(HMG)/Myotubes-Sox6-ChIP-Seq(GSE32627)/Homer	1e-54
6		Sox15(HMG)/CPA-Sox15-ChIP-Seq(GSE62909)/Homer	1e-51
7		Sox17(HMG)/Endoderm-Sox17-ChIP-Seq(GSE61475)/Homer	1e-38
8		Sox4(HMG)/proB-Sox4-ChIP-Seq(GSE50066)/Homer	1e-36
9		Sox9(HMG)/Limb-SOX9-ChIP-Seq(GSE73225)/Homer	1e-33
10		Sox7(HMG)/ESC-Sox7-ChIP-Seq(GSE133899)/Homer	1e-31
72		PBX2(Homeobox)/K562-PBX2-ChIP-Seq(Encode)/Homer	1e-3
73		NF1(CTF)/LNCAP-NF1-ChIP-Seq(Unpublished)/Homer	1e-3
74		JunD(bZIP)/K562-JunD-ChIP-Seq/Homer	1e-3

The HOMER motif results of the gain 1341 binding sites upon NFIA knockdown, we didn't observe NF1 motif.

Rank	Motif	Name	P-value
1		Oct4:Sox17(POU,Homeobox,HMG)/F9-Sox17-ChIP-Seq(GSE44553)/Homer	1e-6
2		FosI2(bZIP)/3T3L1-FosI2-ChIP-Seq(GSE56872)/Homer	1e-6
3		Jun-AP1(bZIP)/K562-cJun-ChIP-Seq(GSE31477)/Homer	1e-6
4		Sox2(HMG)/mES-Sox2-ChIP-Seq(GSE11431)/Homer	1e-5
5		Sox21(HMG)/ESC-SOX21-ChIP-Seq(GSE110505)/Homer	1e-5
6		Sox7(HMG)/ESC-Sox7-ChIP-Seq(GSE133899)/Homer	1e-5
7		JunB(bZIP)/DendriticCells-Junb-ChIP-Seq(GSE36099)/Homer	1e-5
8		Sox15(HMG)/CPA-Sox15-ChIP-Seq(GSE62909)/Homer	1e-5
9		Fra2(bZIP)/Striatum-Fra2-ChIP-Seq(GSE43429)/Homer	1e-4
10		Sox17(HMG)/Endoderm-Sox17-ChIP-Seq(GSE61475)/Homer	1e-4

In the experiment 2, upon the NFIA knockdown, among the lost (3720) and common (2616) SOX2 binding sites, we observed significant enrichment of NFI binding motif. Among the

gained (1143) peaks, we observe less significant enrichment of NFI binding motif. The analysis results shown as below

The HOMER motif results of the lost 3720 SOX2 binding sites upon NFIA knockdown:

Rank	Motif	Name	P-value
1		Sox3(HMG)/NPC-Sox3-ChIP-Seq(GSE33059)/Homer	1e-396
2		Sox21(HMG)ESC-SOX21-ChIP-Seq(GSE110505)/Homer	1e-360
3		Sox2(HMG)/mES-Sox2-ChIP-Seq(GSE11431)/Homer	1e-333
4		Sox10(HMG)SciaticNerve-Sox3-ChIP-Seq(GSE35132)/Homer	1e-323
5		Sox6(HMG)Myotubes-Sox6-ChIP-Seq(GSE32627)/Homer	1e-297
6		Sox15(HMG)/CPA-Sox15-ChIP-Seq(GSE62909)/Homer	1e-292
7		Sox17(HMG)Endoderm-Sox17-ChIP-Seq(GSE61475)/Homer	1e-265
8		Fos12(bZIP)/3T3L1-Fos12-ChIP-Seq(GSE56872)/Homer	1e-226
9		Fra2(bZIP)/Striatum-Fra2-ChIP-Seq(GSE43429)/Homer	1e-225
10		Jun-AP1(bZIP)/K562-cJun-ChIP-Seq(GSE31477)/Homer	1e-221
11		Fra1(bZIP)/BT549-Fra1-ChIP-Seq(GSE46166)/Homer	1e-212
12		Sox4(HMG)/proB-Sox4-ChIP-Seq(GSE50066)/Homer	1e-211
13		Fos(bZIP)/TSC-Fos-ChIP-Seq(GSE110950)/Homer	1e-208
14		JunB(bZIP)/DendriticCells-Junb-ChIP-Seq(GSE36099)/Homer	1e-206
15		Atf3(bZIP)/GBM-ATF3-ChIP-Seq(GSE33912)/Homer	1e-197
16		BATF(bZIP)/Th17-BATF-ChIP-Seq(GSE39756)/Homer	1e-189
17		AP-1(bZIP)/ThioMac-PU.1-ChIP-Seq(GSE21512)/Homer	1e-174
18		Sox7(HMG)ESC-Sox7-ChIP-Seq(GSE133899)/Homer	1e-158
19		Sox9(HMG)/Limb-SOX9-ChIP-Seq(GSE73225)/Homer	1e-98
20		Bach2(bZIP)/OCLY7-Bach2-ChIP-Seq(GSE44420)/Homer	1e-97
21		CDX4(Homeobox)/ZebrafishEmbryos-Cdx4.Myc-ChIP-Seq(GSE48254)/Homer	1e-80
22		HOXB13(Homeobox)/ProstateTumor-HOXB13-ChIP-Seq(GSE56288)/Homer	1e-72
23		Hoxa11(Homeobox)/ChickenMSG-Hoxa11.Flag-ChIP-Seq(GSE86088)/Homer	1e-68
24		NF1(CTF)/LNCAP-NF1-ChIP-Seq(Unpublished)/Homer	1e-60

The HOMER motif results of the 2616 common binding sites:

Rank	Motif	Name	P-value
1		Sox3(HMG)/NPC-Sox3-ChIP-Seq(GSE33059)/Homer	1e-508
2		Sox21(HMG)/ESC-SOX21-ChIP-Seq(GSE110505)/Homer	1e-451
3		Sox2(HMG)/mES-Sox2-ChIP-Seq(GSE11431)/Homer	1e-446
4		Sox10(HMG)/SciaticNerve-Sox3-ChIP-Seq(GSE35132)/Homer	1e-423
5		Sox6(HMG)/Myotubes-Sox6-ChIP-Seq(GSE32627)/Homer	1e-394
6		Sox15(HMG)/CPA-Sox15-ChIP-Seq(GSE62909)/Homer	1e-390
7		Sox17(HMG)/Endoderm-Sox17-ChIP-Seq(GSE61475)/Homer	1e-349
8		Sox4(HMG)/proB-Sox4-ChIP-Seq(GSE50066)/Homer	1e-275
9		Sox7(HMG)/ESC-Sox7-ChIP-Seq(GSE133899)/Homer	1e-210
10		Sox9(HMG)/Limb-SOX9-ChIP-Seq(GSE73225)/Homer	1e-159
62		Hoxd12(Homeobox)/ChickenMSG-Hoxd12.Flag-ChIP-Seq(GSE86088)/Homer	1e-21
63		NF1(CTF)/LNCAP-NF1-ChIP-Seq(Unpublished)/Homer	1e-19
64		CREB5(bZIP)/LNCaP-CREB5.V5-ChIP-Seq(GSE137775)/Homer	1e-19

The HOMER motif results of the gain 1143 binding sites upon NF1A knockdown:

Rank	Motif	Name	P-value
1		Sox3(HMG)/NPC-Sox3-ChIP-Seq(GSE33059)/Homer	1e-156
2		Sox21(HMG)/ESC-SOX21-ChIP-Seq(GSE110505)/Homer	1e-148
3		Sox2(HMG)/mES-Sox2-ChIP-Seq(GSE11431)/Homer	1e-145
4		Sox10(HMG)/SciaticNerve-Sox3-ChIP-Seq(GSE35132)/Homer	1e-130
5		Sox17(HMG)/Endoderm-Sox17-ChIP-Seq(GSE61475)/Homer	1e-111
6		Sox6(HMG)/Myotubes-Sox6-ChIP-Seq(GSE32627)/Homer	1e-110
7		Sox15(HMG)/CPA-Sox15-ChIP-Seq(GSE62909)/Homer	1e-107
8		Sox4(HMG)/proB-Sox4-ChIP-Seq(GSE50066)/Homer	1e-79
9		Sox7(HMG)/ESC-Sox7-ChIP-Seq(GSE133899)/Homer	1e-75
10		Fra1(bZIP)/BT549-Fra1-ChIP-Seq(GSE46166)/Homer	1e-55
96		AMYB(HTH)/Testes-AMYB-ChIP-Seq(GSE44588)/Homer	1e-3
97		NF1(CTF)/LNCAP-NF1-ChIP-Seq(Unpublished)/Homer	1e-3
98		Meis1(Homeobox)/MastCells-Meis1-ChIP-Seq(GSE48085)/Homer	1e-3

3) Using luciferase-reporter assays the authors examine how TF function is affected by the loss of NFIA. Are the regulatory DNA regions containing NFI binding motifs? If so, what is the distance between binding sites of the examined TFs? Most TFs appear to be affected by the loss of NFIA, even TFs that are not interacting with NFIA. The authors conclude that this is due to an indirect regulation of their activities. Why are not these important controls shown in the main figure? Why are only synthetic regulatory DNA-regions analyzed?

To generate luciferase-reporter assays to assess TFs transcriptional activity or pathway activity, the most common approach utilizes the generation of concatenated repeats for a specific transcriptional response elements (TREs) specific for each TF. Most common number of the TREs is 8 or 16. In this study to generate the large number of the TF reporters we also employed this strategy as this is almost the only option to obtain over 2-fold induction levels. Majority of the large 1kb promoter regions give really low activation >2 - fold and are not really sensitive nor stable enough to robustly detect the TF or pathway activation. Of note is that all of the reporters used in this study had minimally 5-fold activation the presence of the transfected corresponding TF. Also in large scale testing no cross-activation with other TFs was observed. This indicates that the NFI function is mediated by interactions and/or recruitment of other transcription regulating factor to the site of transcription.

Reviewer #4 (Expertise: MS, BioID-MS of nuclear proteins):

The manuscript entitled “Human transcription factor protein interaction networks” describes an extensive characterization of PPI networks of 110 transcription factors using AP-MS and BioID. The paper thus provides an extensive list of several thousand potential interactions. Two approaches are included. The first one is using the single-step Strep-tag affinity purification for AP-MS and the second one is using the same system but to purify proteins that were biotinylated by BirA* following addition of biotin to the media. It is mentioned that the expression of studied TF was adjusted to close to physiological level of the Tet inducible system, but no experimental evidence is presented in the paper. Considering the low expression level of TF in general, I doubt this was possible to achieve using the Flp-In Tet inducible system, which is notoriously difficult to titrate.

Regarding the large amount of data and the number of PPI identified, I find it relatively troubling that such a large proportion of interactions (over 75%) were not previously reported. Other large-scale studies with TF published have very little overlap with their own, as mentioned in the text. There are also several studies with transcription factors that have been performed using BioID that could have been used to compare with their own BioID, which has identified the most interactions. This is hand-waved on the fact that the difference is likely due to the transient nature of the TF interactions and/or the different tagging strategies used, but in the absence of validation, it is difficult to determine which of the studies (theirs or the others) represent a true catalogue of PPI for TFs.

Overall, the manuscript is interesting and includes a large amount of data. The following analysis performed underlines interesting observations, including the interactions with other transcription factors and the identification of biological complexes. However, the data presented in a large number of tables makes the data somewhat hard to digest.

We thank the reviewer finding our manuscript interesting and including large amount of data. We also acknowledge the fact that the amount of data can be, in part overwhelming, and have now tried to distill the major findings even better to the revised version. For this the four reviewers comments were extremely helpful.

In respect to validation of our identified interaction, we have now added extensive co-IP validation for the NFI family members as suggested by the reviewers. This validation resulted of a large portion 65/70 (93%) of the tested MS detected interaction pairs to be validated, significantly strengthening the validity and biological relevance of our identified interactions. We also have accordingly adjusted our initial claims that the validation of the identified interactions might be difficult with co-IP due to possibly transient nature of the interactions.

Additionally, we have now modified the text regarding the TF expression level from “close to physiological” to from “physiological to mild or moderate overexpression” (line 108) We and others have observed in previous large-scale studies that the transgene expression obtained with Flp-In Tet inducible system often resemble well with the expression endogenous protein (eg. Silke et al. PMID: 16263936, Glatter et al. PMID: 19156129, Varjosalo et al. PMID: 23602568, Varjosalo et al. PMID: 23455922, Go et al. PMID: 34079125 and others), however, in respect to transcription factors the validation of the physiological expression levels proved extremely difficult due to the lack of good quality antibodies against the TFs in reasonable numbers.

Other comments:

The identified proteins were filtered using a SAINT score cut-off of 0.74, but no further discussion is provided on why this value was used. Considering the analysis presented the results from 2-4 repeats for each TF, I think it would be appropriate to provide a better rationale. Is this threshold actually stringent, and are they losing significant proteins that should be taken into account?

The threshold and the updated filtering strategy is very stringent. The SAINT score cut-off of 0.74 alone actually requires that the bait has been seen in minimally $\frac{3}{4}$ samples.

To further improve the filtering we have now added the control purification data from CRAPome-database (that includes contaminant proteins from 716 experiments). In BioID data, only the proteins with lower frequency than 50 % (358/716) in CRAPome were allowed. Also preys with CRAPome frequency 25-50% (179-358/716) and with higher CRAPome average spectral count compared to our average spectral count were removed. This led to deletion of 520 interactions from BioID data (Table S1B).

From AP-MS data, in addition to Saint cut-off, preys that were present with higher frequency than 50% in Crapome (358/716) were required to have three-time higher average spectral count than average spectral count in CRAPome. This led to deletion of 640 interactions from AP-MS data (Table S1C).

As the BioID and AP-MS methods differ from each other in interaction stability, same filtering methods cannot not be utilized as it leads to not stringent enough filtering of BioID data. However, we think that now the strict filtering used, results to a very high-confidence interactions. Tables and figures are now corrected accordingly.

Some important information is missing for protein identification with Proteome Discoverer. An FDR for peptide was set to 0.05. What was the minimum number of peptides used in the study? Was that taken into accounts when identifying an interaction? For example, an identification in at least 2 of the repeats could be used to filter out potential contaminants.

The SAINT statistical filtering does take into consideration the spectral count values of the detected interactions and compares these with the controls. SAINT filtering average spectral count cut-off was set to ≥ 2 (detected in $\geq 3/4$ of the samples). This results to high-confidence interacting proteins to been assigned with the very stringent two-peptide rule. Additionally, we have made the filtering even more stringent than before as discussed on the previous reply above.

The quality of the text is poor. There are several errors that goes beyond typos or grammatical errors. The text will have to be extensively edited for English.

As a non-native English speaker and writers, we deeply apologize for this. We have now used a Nature publishing group's professional scientific language (English) editing service to improve grammar and language. We believe this has significantly improved the quality of the text.

Reviewers' Comments:

Reviewer #1:

Remarks to the Author:

I'm satisfied with the authors' responses to my comments and requests. I think the added filtering the authors have incorporated has enhanced the quality of their datasets and they will be a useful resource for the scientific community.

Reviewer #2:

Remarks to the Author:

In the revised version of the manuscript entitled of "Human transcription factor protein interaction networks", the authors performed additional co-IP validation for a large set of stable interactions with quite good success rate. However, for the main concerns related to data quality and data presentation raised by most of the reviewers, it is disappointed that the authors only addressed quite generally. Without significant improvement for the evaluation of the data quality and data presentation, such type of large-scale resource with great effort is hard to be appreciated by the audience and meet the quality of Nat. Commun. Several examples for these major concerns are listed as below.

1. The authors only performed further comparison to the three papers listed as examples. Systematic survey for most representative works should be done. For example, the recent BioID human cell map (Nature, 2021, 595, 120).
2. The authors highlighted that "the correlation between the technical and biological replicates were excellent (Figure S1A)". What type of data was investigated and presented? Without comprehensive and solid evaluation, such conclusion tends to be overstated. "We thank the reviewer for suggesting adding the peptide counts and other details also in figures. We tried this, but for our and colleagues' opinion this resulted in extremely busy and difficult figures, and we therefore decided to leave these details to table format (Table S1)" These main figures are full of busy hairball maps which provides limited biological information. Experimental data should be provided at least for the representative examples presented in Figure 4 and 6.
3. The authors highlighted the use of SAINT and CRAPome for their data analysis, and therefore, concluded their data presentation is in high quality. The authors should present detailed data at least in Figure 4 and 6 for confirming such a conclusion.
4. For the major BioID interactome, the authors still didn't provide any validation and further solid data analysis. Detailed data analysis workflow should be provided.

Reviewer #3:

Remarks to the Author:

The authors of the manuscript titled "Human transcription factor protein interaction network" have responded to several of the comments that I raised in the initial review. However, I'm still concerned by the fact that the authors only use an ectopic context (293 T-Rex cell line) to examine the functional interaction between lineage specific transcription factors. Moreover, the authors present data that lacks proper controls and that they fail to discuss/interpret.

Specific comments

- 1) The authors show that upon removal of NFIA, the genome wide binding pattern of misexpressed SOX2 is substantially altered in 293-cells. Without further experiments explaining how NFIA affects the binding pattern of SOX2, this data set is inconclusive and can be left out.
- 2) In Fig. S4B the authors present a pathway enrichment analysis of genes targeted by misexpressed SOX2 in 293-cells, both in the presence and absence of NFIA. What is the relevance of this data set. Does SOX2 has a significant role in these cells that can be determined with a pathway enrichment analysis?
- 3) In Fig. 4 and Fig. S5 the authors examine how the loss of NFIA affects the capacity of a set of transcription factors to regulate a reporter gene. Using synthetic enhancers that consists of multimerized specific transcription factor binding sites, the authors show that the loss of NFIA affects the activity of transcription factors, regardless of their interaction with NFIA. Again, without

further experiments giving a reasonable explanation for this result, this data set can be left out.

Reviewer #4:

Remarks to the Author:

The manuscript entitled "Human transcription factor protein interaction networks" is a resubmission following extensive corrections and additions.

I believe the authors have made considerable efforts to address the comments from all reviewers, including more comparison with other large-scale studies of PPI of transcription factors.

Moreover, the validation of NFI family members interactions provide a very strong argument for the quality of their data.

Additional details on filtering and scoring of the interactions are also explained properly.

Finally, the text has now been edited to proper standards.

I believe the article is now acceptable.

REVIEWERS' COMMENTS

Reviewer #1 (Remarks to the Author):

I'm satisfied with the authors' responses to my comments and requests. I think the added filtering the authors have incorporated has enhanced the quality of their datasets and they will be a useful resource for the scientific community.

Reviewer #2 (Remarks to the Author):

In the revised version of the manuscript entitled of “Human transcription factor protein interaction networks”, the authors performed additional co-IP validation for a large set of stable interactions with quite good success rate. However, for the main concerns related to data quality and data presentation raised by most of the reviewers, it is disappointed that the authors only addressed quite generally. Without significant improvement for the evaluation of the data quality and data presentation, such type of large-scale resource with great effort is hard to be appreciated by the audience and meet the quality of Nat. Commun. Several examples for these major concerns are listed as below.

1. The authors only performed further comparison to the three papers listed as examples. Systematic survey for most representative works should be done. For example, the recent BioID human cell map (Nature, 2021, 595, 120).

We thank reviewer for the comment. The editorial team agrees that no further systematic comparisons is needed. In our opinion the comparison to public interaction databases such as PINA2, STRING, IntAct and BioGRID and several medium- to large-scale interactome studies such as Li et al., Lambert et al., Malovannaya et al., and Huttlin et al. provides already a comprehensive overview of overlapping interactions. In addition, we show that while the overlap of identified interactions between our dataset and the previously mentioned large-scale interactomes studies is low, so is the overlap among the previous studies as well. Indeed, our results overlap more with the existing data, than many of the other interactomes (Figure S1B). This may suggest that all the studies capture slightly different facets of TF interactomes, and together form a much more complete picture, than any one study alone.

2. The authors highlighted that “the correlation between the technical and biological replicates were excellent (Figure S1A)”. What type of data was investigated and presented? Without comprehensive and solid evaluation, such conclusion tends to be overstated.

The correlation analysis of technical and biological replicates is shown in Figure S1A. As stated in figure legend “Correlation plots of biological (blue) and technical (orange) replicates performed with seaborn python package”.

We found the correlation extremely high (>0,970 with biological replicates and > 0.974 with technical replicates, both with the p-values near to 0.)

We have now added a more detailed description of the used correlation analysis in methods section: “Correlation for biological and technical replicates was analyzed using spectral count values of either biological or technical replicates. Pearson’s correlation coefficient values were calculated for each

pair of replicates with the personr method from the python scipy.stats package (SciPy, version 1.71). Plots for the results were generated with Implot method of the python seaborn package (version 0.11.2). “

3. The authors highlighted the use of SAINT and CRAPome for their data analysis, and therefore, concluded their data presentation is in high quality. The authors should present detailed data at least in Figure 4 and 6 for confirming such a conclusion.

“We thank the reviewer for suggesting adding the peptide counts and other details also in figures. We tried this, but for our and colleagues’ opinion this resulted in extremely busy and difficult figures, and we therefore decided to leave these details to table format (Table S1)” These main figures are full of busy hairball maps which provides limited biological information. Experimental data should be provided at least for the representative examples presented in Figure 4 and 6.

In the revision as quoted by the reviewer, we agreed to do this if the reviewer thinks this is important for him/her. As per request from the Reviewer the quantitative experimental data on the interaction abundances has now been added to Figure 4 and Figure 6.

4. For the major BioID interactome, the authors still didn’t provide any validation and further solid data analysis. Detailed data analysis workflow should be provided.

We cordially disagree with the reviewer on NOT providing ANY validation and SOLID data analyses on the BioID interactions. In fact, in the revision, we added validation of almost hundred protein-protein interactions. To our knowledge the number of interactions validated with a complementary analysis in this study compared to other interaction proteomics studies, is in the top 5%.

This criticism is also somewhat disagreement with the other comments the revised manuscript obtained:

“I believe the authors have made considerable efforts to address the comments from all reviewers, including more comparison with other large-scale studies of PPI of transcription factors.”

“Moreover, the validation of NFI family members interactions provide a very strong argument for the quality of their data.”

“Additional details on filtering and scoring of the interactions are also explained properly.”

Reviewer #3 (Remarks to the Author):

The authors of the manuscript titled “Human transcription factor protein interaction network” have responded to several of the comments that I raised in the initial review. However, I’m still concerned by the fact that the authors only use an ectopic context (293 T-Rex cell line) to examine the functional interaction between lineage specific transcription factors. Moreover, the authors present data that lacks proper controls and that they fail to discuss/interpret.

Specific comments

- 1) The authors show that upon removal of NFIA, the genome wide binding pattern of misexpressed SOX2 is substantially altered in 293-cells. Without further experiments explaining how NFIA affects the binding pattern of SOX2, this data set is inconclusive and can be left out.
- 2) In Fig. S4B the authors present a pathway enrichment analysis of genes targeted by misexpressed SOX2 in 293-cells, both in the presence and absence of NFIA. What is the relevance of this data set. Does SOX2 has a significant role in these cells that can be determined with a pathway enrichment analysis?

As suggested by the Reviewer and agreed by the editorial board, we have now removed the NFIA and Sox2 binding data and modified the figures and text accordingly.

- 3) In Fig. 4 and Fig. S5 the authors examine how the loss of NFIA affects the capacity of a set of transcription factors to regulate a reporter gene. Using synthetic enhancers that consists of multimerized specific transcription factor binding sites, the authors show that the loss of NFIA affects the activity of transcription factors, regardless of their interaction with NFIA. Again, without further experiments giving a reasonable explanation for this result, this data set can be left out.

As suggested by the Reviewer, we agree to remove the Figure S5. Although interesting observation, finding and validating a clear indirect mechanism of action of NFIA regulation of other transcription factor requires further studies. We have now highlighted in the text that measured transcriptional activity of each single TF alone and the effect of NFIA interaction on these:” Of note is that each of these assays measure only the possible binding activity change of each specific TF. TFs bind to several different sites in the DNA (Jolma et al., 2013) and it is highly unlikely that DNA mediated interactions would be detected in these abundancies as detected by our interactome analyses.” (rows 335-338).

Jolma, A. et al. DNA-binding specificities of human transcription factors. *Cell*. 152, 327-339 (2013).

Reviewer #4 (Remarks to the Author):

The manuscript entitled “Human transcription factor protein interaction networks” is a resubmission following extensive corrections and additions.

I believe the authors have made considerable efforts to address the comments from all reviewers, including more comparison with other large-scale studies of PPI of transcription factors.

Moreover, the validation of NFI family members interactions provide a very strong argument for the quality of their data.

Additional details on filtering and scoring of the interactions are also explained properly.

Finally, the text has now been edited to proper standards.

I believe the article is now acceptable.